# Linking the Amyloid, Tau, and Mitochondrial Hypotheses of Alzheimer’s Disease and Identifying Promising Drug Targets

**DOI:** 10.3390/biom12111676

**Published:** 2022-11-11

**Authors:** Zdeněk Fišar

**Affiliations:** Department of Psychiatry, First Faculty of Medicine, Charles University and General University Hospital in Prague, Ke Karlovu 11, 120 00 Prague, Czech Republic; zfisar@lf1.cuni.cz; Tel.: +420-224965313

**Keywords:** Alzheimer’s disease, amyloid beta, mitochondria, tau protein, drug

## Abstract

Damage or loss of brain cells and impaired neurochemistry, neurogenesis, and synaptic and nonsynaptic plasticity of the brain lead to dementia in neurodegenerative diseases, such as Alzheimer’s disease (AD). Injury to synapses and neurons and accumulation of extracellular amyloid plaques and intracellular neurofibrillary tangles are considered the main morphological and neuropathological features of AD. Age, genetic and epigenetic factors, environmental stressors, and lifestyle contribute to the risk of AD onset and progression. These risk factors are associated with structural and functional changes in the brain, leading to cognitive decline. Biomarkers of AD reflect or cause specific changes in brain function, especially changes in pathways associated with neurotransmission, neuroinflammation, bioenergetics, apoptosis, and oxidative and nitrosative stress. Even in the initial stages, AD is associated with Aβ neurotoxicity, mitochondrial dysfunction, and tau neurotoxicity. The integrative amyloid-tau-mitochondrial hypothesis assumes that the primary cause of AD is the neurotoxicity of Aβ oligomers and tau oligomers, mitochondrial dysfunction, and their mutual synergy. For the development of new efficient AD drugs, targeting the elimination of neurotoxicity, mutual potentiation of effects, and unwanted protein interactions of risk factors and biomarkers (mainly Aβ oligomers, tau oligomers, and mitochondrial dysfunction) in the early stage of the disease seems promising.

## 1. Introduction

Alzheimer’s disease (AD) is a progressive neurodegenerative disease, and the most common cause of AD dementia is neuronal death and loss of synapses in certain areas of the brain. AD has been defined as a clinical dementia syndrome confirmed at autopsy or *in vivo* by the neuropathological observation of neuritic plaques composed of amyloid beta (Aβ) and neurofibrillary tangles (NFTs) composed of paired helical filaments of hyperphosphorylated tau protein (tau). Other typical abnormalities in the brains of AD patients include neuronal loss, synaptic alterations, and neuroinflammation [1]. Critical roles in AD pathogenesis have been assigned to Aβ metabolism [2], tauopathy [3], mitochondrial dysfunction [4], and oxidative stress [5,6], among other pathways.

At the cellular, subcellular, and biochemical levels, AD can be viewed as a proteinopathy with an excessive accumulation of extracellular densely packed Aβ filaments [7] and intraneuronal NFTs [3] in the brain. Soluble Aβ and tau oligomers are responsible for neuronal damage and death by disrupting brain chemistry and synaptic plasticity [8,9,10,11,12,13,14]; thus, AD can also be viewed as and oligomeropathy [15]. Soluble Aβ and tau oligomers can propagate via prion-like mechanisms in the brain [16]. Aβ oligomers trigger the conversion of tau to the toxic oligomeric form; at the same time, through a feedback loop, tau can increase in the toxicity of Aβ oligomers [17]. Neuroinflammation [18], oxidative stress [19,20], and disruption of metabolic pathways [21,22] are involved in the complex cascade leading to AD pathology and symptoms. Reactive oxygen species (ROS) overproduction is thought to play a critical role in both the accumulation and deposition of Aβ [23], tau hyperphosphorylation, and intracellular NFT formation [24] in AD. Mitochondrial dysfunction leading to impaired bioenergetics and calcium homeostasis, oxidative stress, membrane depolarization and permeabilization, and apoptosis plays a key role in the initiation and/or regulation of all cellular processes, including the neurotoxicity of Aβ and tau [25]. AD biomarkers include genetic, biochemical, physiological, and neuroimaging parameters involved in neurodegeneration, impaired neuroplasticity, and brain structure and function damage.

AD is classified as an Aβ and tau pathology leading to neurodegeneration and cognitive impairment [26,27]. The onset of AD and disease progression are determined by a combination of environmental and genetic risk factors and biomarkers, resulting in disturbances in various brain areas and neurocircuits, neurodegeneration, and impaired neuroplasticity and neurochemistry, which are responsible for disease symptoms (Figure 1).

Most cases of AD are recognized after age 65 and are called sporadic late-onset AD (LOAD). In 5–10% of cases there is an earlier onset of the disease; this form of AD is referred to as familial early onset AD (EOAD) and is strongly genetically determined. Autosomal dominant AD (ADAD) is defined as dominantly inherited AD with pathological confirmation. ADAD occurs in less than 1% of all cases in which dementia develops at a predictable age, due to specific genetic mutations. The study of ADAD makes it possible to determine the sequence of changes in biomarkers in persons destined to develop genetically conditioned AD [28,29]. For LOAD, this sequence may or may not be the same. ADAD, EOAD, and LOAD are thought to share similar pathophysiological features.

Currently, there is only symptomatic treatment for AD [30]. Progress in understanding the regulatory processes underlying neurodegeneration and finding specific and sensitive biomarkers of prodromal and early stage of AD are necessary to enable diagnosis of incipient AD and the development of disease-modifying (causal) drugs for initial treatment of the disease.

This review summarizes findings on risk factors and biomarkers of AD that are associated with changes in brain chemistry and neuroplasticity leading to neurodegeneration. The role of Aβ, tau, and mitochondria in the development of AD is presented in terms of the search for primary causes and triggers of AD and their mutual interactions leading to neurodegeneration. The integrative amyloid-tau-mitochondrial hypothesis of AD captures the interconnection of the main causes of neurodegeneration in AD and reflects the time course of biomarkers, mitochondrial targets of Aβ and tau, and the complex and multifactorial nature of AD. Finally, the cellular targets of AD drugs are summarized, particularly candidate drugs developed to eliminate neurodegeneration in the early stage of the disease. Article search was conducted primarily using PubMed with searches based on combinations of keywords “Alzheimer’s disease”, “amyloid beta”, “mitochondria”, “tau”, and “drug”.

## 2. Risk Factors and Biomarkers of Alzheimer’s Disease

The terms risk factor, risk marker, and biological marker were defined by the Biomarker Definition Working Group [31]. Risk factors are involved in the causal pathway leading to the disease. Risk markers may not be causally linked to the disease, i.e., they can identify changes associated with the disease process itself. Biological markers (biomarkers) are objectively measured and evaluated as indicators of normal biological processes, pathogenic processes, or pharmacological responses to a therapeutic intervention. Studying risk factors, risk markers, and biomarkers and validating the biological hypotheses of AD are essential for understanding the etiology of this neurodegenerative disease and for finding effective drugs. It is obvious that different risk factors and biomarkers and their interactions may converge and result in the same neurodegenerative changes in AD. Moreover, many environmental factors influence the progression of AD, including lifestyle and cognitive reserve [32,33], indicating that the development of the disease may be highly individual.

### 2.1. Risk Factors

Risk factors for the development of AD dementia include advanced age; female sex [34,35], presence of the APOE ε4 allele, encoding isoform apolipoprotein 4 (ApoE4); other genetic and epigenetic variations [36,37,38]; brain injury [39]; and environmental factors and stressors [40,41], including a low level of education [42,43] and other lifestyle factors [44]; infections [45]; cardiovascular disease [46,47]; type 2 diabetes mellitus [48,49]; and other comorbid metabolic diseases [50].

The percentage of people with AD increases dramatically with age; therefore, the most prominent risk factor for AD is age. AD is defined by a specific neuropathology (amyloid plaques and NFTs) and is therefore not an accelerated aging of the brain [51]. However, specific AD neuropathology may be triggered or accelerated by certain processes that accompany normal aging, such as mitochondrial dysfunction [52,53,54,55]. More women than men suffer from AD and other dementias. Differences in dementia risk between men and women may depend on age, geographic region, hormonal changes, and other factors [56,57,58].

#### 2.1.1. Environmental

Environmental stressors/risks contributing to the development of dementia include traumatic brain injury, ischemia, hypoxia, neurotoxins, infections, lifestyle (drug abuse, exercise, and education), and metabolic and cardiovascular comorbid diseases (blood pressure, cholesterol, obesity, diabetes, depression, etc.) [59]. Modifiable risk factors for AD dementia include cardiovascular disease [60], physical activity [61], diet [62], education [63], traumatic brain injury [64,65], depression [66] and others. However, evidence that environmental influences contribute to specific pathological changes in AD is limited.

An umbrella review of systematic reviews and meta-analyses evaluating the association between environmental risk factors and AD showed that late-life depression and type 2 diabetes mellitus are the most significant; cancer, depression at any age, and physical activity are also highly significant (*p* < 10^−6^) in AD development. Less significant (*p* < 10^−3^) is aluminum, education, herpesviridae infection, and low-frequency electromagnetic fields, and NSAIDs. Weak significance (*p* < 0.05) was observed for alcohol consumption, dietary intake of vitamin C, dietary intake of vitamin E, chlamydia pneumoniae infection, spirochetal infection, obesity, mild traumatic brain injury, statins, agreeableness, conscientiousness, neuroticism, openness, aspirin, nonaspirin NSAIDs, fish intake, and stroke [40]. Weakly significant were also agreeableness, conscientiousness, and openness [67].

#### 2.1.2. Genetic

The estimated heritability of AD is approximately 70%; thus, genetic variation is a significant contributor to the risk of the disease [68]. Genetic risk factors and biomarkers are particularly decisive in familial EOAD, where mutations in genes for amyloid precursor protein are involved (*APP*), presenilin 1 (*PSEN1*), and presenilin 2 (*PSEN2*) are involved [69]. The most common cause of EOAD inheritance is *PSEN1* mutation, *PSEN2* and *APP* mutations are less common [70]. These mutations lead to overproduction of Aβ_42_ or an altered Aβ_42_/Aβ_40_ concentration ratio and amyloid neurotoxicity in AD.

The presence of the *APOE* ɛ4 allele is a major genetic risk factor for sporadic LOAD [38,71]. ApoE4-enhanced metabolic changes and neuroinflammation via amyloid toxicity, tau toxicity, mitochondrial dysfunction, mitophagy, and insulin resistance may contribute to neurodegenerative changes in AD [72,73]. ApoE4 enhances brain Aβ pathology, tau pathogenesis, and tau-mediated neuroinflammation and neurodegeneration independent of Aβ pathology [74]. A model has been proposed in which the pathological effects of Aβ and tau leading to neurodegeneration are enhanced in the presence of ApoE4. According to this model, ApoE4 amplifies the pathogenesis and neurotoxicity of Aβ and tau, leading to (1) ApoE4-enhanced neuronal damage and death, induction of neuroinflammation that may also be enhanced by ApoE4, and further neurodegeneration; (2) neuronal death enhanced by ApoE4 and subsequent induction of neuroinflammation, which further enhances neurodegeneration; and (3) ApoE4-enhanced neuroinflammation, which induces increased neurodegeneration and neuronal death.

In addition to the *APOE* locus, GWAS and next-generation sequencing methods have identified approximately 40 susceptibility loci associated with AD [75,76,77,78,79]. The results showed that disturbed APP metabolism is associated not only with EOAD, but also with LOAD. In addition, tau was found to potentially play a role in early AD pathology [74,80], and immune and lipid metabolism pathways may be involved. The study of rare genetic variants associated with AD can identify novel loci, located, e.g., within *PLD3*, *TREM2*, *ABI3*, *PLCG2*, *PILRA*, *ABCA7*, and *SORL1* [36,81,82]. Genomic risk loci need to be mapped to variants and genes through functional genomics studies.

Epigenetic changes (methylation and hydroxy methylation of DNA, histone acetylation and methylation, and noncoding RNAs regulation) have been shown to be important in the pathogenesis of AD. Even mtDNA can be epigenetically regulated, e.g., by DNA methylation and noncoding RNAs. Given that epigenetic and mitoepigenetic changes can be detected even in the periphery, they have the potential to be included in the list of AD risk factors and biomarkers and to become therapeutic targets for new drugs. The epigenetics of AD, including epigenetic modulation by microRNAs, and possible treatment opportunities have been described and discussed in detail in a number of reviews [83,84,85,86,87,88,89].

### 2.2. Biomarkers

AD is defined by pathological processes that can be documented by biomarkers measured postmortem or *in vivo*. Due to the difficulty of clinical diagnosis of AD, determining the prognosis of the disease, and monitoring of treatment results, biomarkers have an important role in this field [26]. Biomarkers serve as diagnostic indicators or as markers of pathological changes in the initial stages of the disease. The currently employed diagnostic methods are frequently costly, and time-consuming, invasive, poorly accessible, and insufficiently sensitive for detection of the initial stages of the disease. Specific, sensitive, noninvasive, and inexpensive biomarkers of AD capable of identifying early onset disease are essential for screening in the preclinical stage, which is crucial for the current as well as the future therapy [90]. The most promising biomarkers are genetic, biochemical (measured in cerebrospinal fluid (CSF) or in peripheral blood) and derived from structural and functional neuroimaging [91].

The brain processes, in which the production and effects of Alzheimer’s biomarkers occur, are amyloid pathology (toxicity of amyloid oligomers and plaques), tau pathology (toxicity of tau oligomers and tangles), mitochondrial dysfunction, neuroinflammation, neurodegeneration, synaptic changes, and blood–brain barrier alterations. Knowing the temporal development of biomarkers is important for understanding the etiology of AD, monitoring disease progression and targeted treatment of the disease. The AT(N) system, a classification scheme based on measuring biomarkers of amyloid pathology, tau pathology, and neurodegeneration, is proposed for clinical practice [92]. This system can be further expanded to include mitochondrial, inflammatory, synaptic, oxidative stress, and vascular biomarkers [93].

#### 2.2.1. CSF and Neuroimaging

The long-accepted biochemical biomarkers of AD are low CSF Aβ_42_ concentrations, reflecting brain Aβ deposition, and markers of neuronal degeneration or injury, such as elevated CSF tau. Tau biomarkers include both total tau (T-tau) and phosphorylated tau (P-tau). The Erlangen Score algorithm can be used to facilitate the clinical interpretation of CSF Aβ and tau AD biomarkers [94].

Current biomarkers (Aβ, tau, and neurodegeneration) are measured in CSF or by brain neuroimaging. Specific Aβ-PET and tau-PET tracers enable the study of the misfolded and accumulated proteins (but not monomers and oligomers) in the brain *in vivo* [95,96,97,98,99]. In the brain, Aβ deposition may be monitored by ^11^C-Pittsburgh compound B (^11^C-PiB) or ^18^F-florbetapir PET; tau deposition can be monitored using newer ^18^F-based tau binding ligands; and neurodegeneration and synaptic damage can be observed via ^18^F-fluorodeoxyglucose (^18^F-FDG) PET or structural magnetic resonance imaging (MRI) data analysis [27,100,101]. Promising neuroimaging biomarkers for AD include hippocampal and entorhinal cortex volume, basal forebrain nuclei, cortical thickness, and others [27,98].

#### 2.2.2. Blood-Based

Blood-based biomarkers represent a less invasive and potentially cheaper approach for aiding AD detection than CSF and neuroimaging biomarkers. Highly sensitive and specific assays have shown reduced plasma Aβ_42_ and Aβ_40_ levels in patients with AD that correlated with both CSF concentrations and PET-assessed Aβ plaque positivity, confirming the potential clinical utility of plasma biomarkers in predicting the individual brain Aβ load [102,103]. Plasma P-tau217, P-tau181, and the Aβ_42_/Aβ_40_ ratio appear to be suitable blood-based biomarkers [103,104,105]. The applicability of AD blood biomarkers assumes that changes in parameters measured in peripheral blood components reflect specific pathological processes in the brain. This approach is supported by the finding that brain-derived exosomes can transport a number of biologically active molecules (proteins, lipids, DNA, and RNAs) across the blood–brain barrier [106]. Ultrasensitive techniques for measuring blood biomarkers for clinical use are developed primarily in the field of immunoassay and mass spectrometry [107].

Spectroscopic techniques also provide complex information about blood-based samples, which makes them well suited for studying multifactorial diseases, such as AD. Changes in the concentrations and structure of biomolecules within the sample are reflected in the overall spectral pattern indicating the presence of the disease. Most of the published studies to date have utilized infrared spectroscopy [108]. Given that spectroscopic methods allow for a rapid, cost-efficient, non-invasive, and reagent-free analysis of plasma molecules without the need for a sophisticated sample preparation, they may serve as a screening tool for the at-risk population.

Numerous proteomic and metabolomics studies [109,110] have been conducted, resulting in panels of biomarkers, which enable the differentiation of AD patients from nondemented controls or even other types of dementia. For example, a panel of potential AD-specific plasma metabolites including arachidonic acid was proposed, *N*,*N*-dimethylglycine, thymine, glutamine, glutamic acid, and cytidine, was proposed [110].

Glycogen synthase kinase-3 beta (GSK-3), cAMP-response element-binding protein (CREB) and brain-derived neurotrophic factor (BDNF) play important roles in neuronal survival, synaptic plasticity, and AD pathophysiology. In AD with depressive symptoms, increased CREB activity in lymphocytes and increased GSK-3 activity in platelets were observed, while in AD without depression, a reduced concentration of BDNF in platelet-rich plasma was confirmed [111]. Thus, comorbid diseases must be considered when interpreting biomarkers measured in peripheral blood. Currently, new measurable markers in peripheral blood are being sought using genomic, epigenomic, proteomic, metabolomic, and lipidomic methods [83,112,113]. The search for these markers is based on the knowledge that, in addition to amyloid and tau proteinopathy, the pathophysiology of AD includes disturbed synaptic homeostasis, impaired lipid and energy metabolism, oxidative stress, and neuroinflammation.

Mitochondrial dysfunction and oxidative stress are significant participants in the intrinsic mechanisms of nerve cell damage leading to neurodegenerative diseases. Therefore, it is necessary to investigate changes in the production of free radicals and in the activity of the antioxidant system. Biomarkers of oxidative stress in the AD brain have been well documented, with markers of protein, lipid, DNA and RNA oxidation [114]. Mitochondrial AD biomarkers include changes in mtDNA, ATP production, ROS production, Ca^2+^ levels, apoptotic factors, and the oxidative phosphorylation (OXPHOS) system. For example, in the peripheral blood, platelets of AD patients have been identified to have decreased complex IV activity, decreased mitochondrial basal respiration, and decreased maximal capacity of electron transport system [115,116]. Various cytokines have been implicated in the pathogenesis of AD, including interleukins (IL-1, IL-6, IL-12, and other families), tumor necrosis factor alfa (TNF-α), transforming growth factor beta (TGF-β), and interferon gamma (IFN-γ), which may serve as biomarkers and therapeutic targets [117].

Potential novel AD biomarkers include neurogranin and visinin-like protein 1 (VILIP-1) (promising synaptic and neuronal degeneration biomarkers of AD) [105,118], neurofilament light polypeptide (NfL, a possible tau-independent marker of neuroaxonal degeneration) [107], glial fibrillary acidic protein (a promising plasma biomarker of AD) [119,120], and soluble triggering receptor expressed on myeloid cells 2 (TREM2) (a possible biomarker of microglial activation) to which ApoE can bind [121,122]. A multistep neurodiagnostic process starting with the examination of blood biomarkers of AD and other neurodegenerative diseases has been explored and discussed [123].

In summary, two biomarkers of amyloidosis (reduced CSF Aβ_42_ in CSF and PET brain amyloid imaging) and biomarkers of neurodegeneration (increased T-tau and P-tau, NfL, and VILIP-1 in CSF, atrophy on structural MRI, and hypometabolism measured by FDG PET) are sufficiently validated for inclusion in the clinical diagnosis of AD and monitoring the effects of therapy [124,125]. A comprehensive systematic review and meta-analysis showed the following AD biomarkers to be promising: (i) in CSF Aβ_42_, T-tau, P-tau, and tau/Aβ_42_; (ii) in peripheral blood T-tau, Aβ_42_/Aβ_40_, and NfL; (iii) in the whole brain, left and right hippocampal volume, entorhinal cortex volume, medial temporal lobe atrophy, ^18^F-FDG PET, ^11^C-PiB PET, and (iv) *APOE* ε4 [126]. Blood-based biomarkers are currently being assessed, with the aim of improving AD diagnosis, predicting cognitive decline, and finding new molecular targets for AD drugs [104,111,113,116,127,128,129,130,131]. The Alzheimer’s Association recommends the use of blood-based markers as (pre-)screeners for preliminary identification of AD, before confirmation of the disease by CSF or PET testing, and for studying longitudinal changes in interventional trials [132].

### 2.3. Time Course of Biomarkers

The temporal order in which nongenetic biomarkers are detectable can help determine the primary cause of AD development. A hypothetical time course involving the neurotoxicity of Aβ and tau oligomers and mitochondrial dysfunction (Figure 2) can be used as a basis for the development of new AD drugs targeting pathophysiological processes in the initial stages of the disease. Given the interconnectedness of brain processes leading to neurodegeneration (e.g., mitochondrial dysfunction, amyloid neurotoxicity, and tau neurotoxicity) and the fact that the onset of LOAD can be 20 or more years before the appearance of symptoms, it makes sense to apply pharmacotherapy to target multiple systems that regulate neurodegeneration, neuroplasticity, and brain neurochemistry in AD.

**Figure 2 biomolecules-12-01676-f002:**
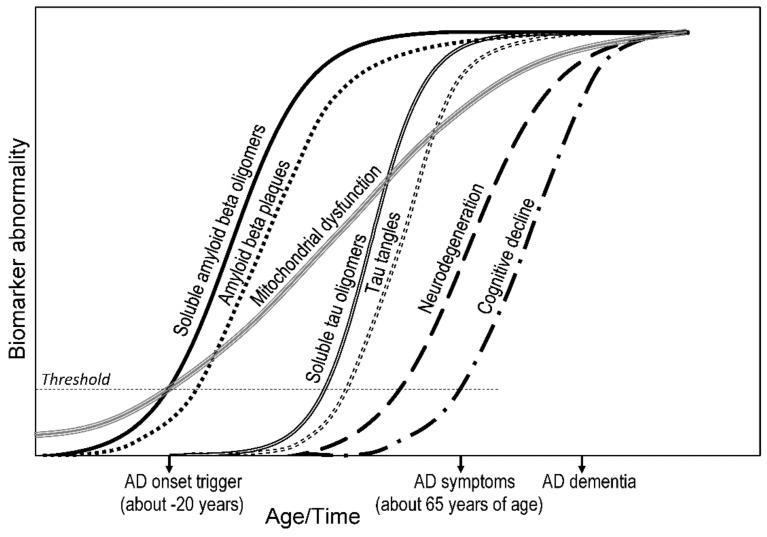
Hypothetical time course of biomarker abnormalities and pathological changes in Alzheimer’s disease (AD) based on (i) measurement of amyloid beta (Aβ) and phosphorylated tau (P-tau) in cerebrospinal fluid and/or brain via PET, (ii) measurement of neurodegeneration via FDG PET and MRI, (iii) assumption of mitochondrial toxicity of both Aβ and tau oligomers and/or mitochondria-induced Aβ or tau pathology, (iv) synaptic dysfunction quantified by FDG-PET and fMRI [95,133], and (v) assumed age-dependent mitochondrial dysfunction. AD symptoms first appear in mid-60s and starting of AD onset can occur more than two decades before that; mitochondrial dysfunction develops gradually in the aging process. Age-dependent threshold crossings of mitochondrial dysfunction, Aβ neurotoxicity, and/or tau neurotoxicity appear to be primary initiator of AD onset, leading to neurodegeneration, symptom onset, and AD dementia. FDG—^18^F-fluorodeoxyglucose; fMRI—functional magnetic resonance imaging; PET—positron emission tomography.

#### 2.3.1. Clinical View

Pathobiological models of AD agree that Aβ and tau pathology accumulates over two decades without symptoms [29,133,134]. While the identification of AD in living persons is based on the measurement of biomarkers [26], the postmortem identification of AD is based on staging systems that are able to assess the gradual progression of the pathology. The National Institute on Aging–Alzheimer’s Association (NIA-AA) recommends [135] that the evaluation of neuropathological changes in AD include three parameters: Aβ plaque score based on Thal phases [80,136], NFT stage based on Braak neuropathological staging [137,138], and neuritic plaque score based on Consortium to Establish a Registry for Alzheimer’s Disease (CERAD) recommendations [139]. For example, the Braak staging based on NFTs uses a staging scheme from no NFTs to a stage with NFTs widely distributed in the neocortex and ultimately involving primary motor and sensory areas [137,140]. The [^18^F]MK6240 tau PET imaging has confirmed the suitability of using the Braak tau staging system in living individuals [141].

From a clinical point of view, the AD continuum includes three stages: (1) preclinical AD (asymptomatic cerebral amyloidosis); (2) amyloid positivity + evidence of synaptic dysfunction and/or early neurodegeneration; and (3) amyloid positivity + evidence of neurodegeneration + subtle cognitive decline, with mild cognitive impairment (MCI) due to AD, and dementia due to AD. The stages of dementia due to AD are divided into mild, moderate, and severe AD dementia [41,142].

#### 2.3.2. Research View

For AD, it is hypothesized that processes associated with aging, influenced by certain genetic factors and by environmental and lifestyle factors, may be triggers of specific progressive neurodegeneration. Pathological processes in the brain begin in the preclinical stage, many years before the diagnosis of AD dementia [143]. NIA-AA published research criteria for preclinical AD; a sequence of biomarkers was proposed to predict the risk of progression toward MCI and AD dementia [26,142]. The concept of the preclinical stage of AD is based on the amyloid hypothesis according to which Aβ accumulation caused by age, genetics, and the environment (including cardiovascular risk factors, age-related brain diseases, and the effect of brain and cognitive reserve) precedes synaptic dysfunction, glial activation, tangle formation, neurodegeneration, and neuronal loss, which leads to cognitive decline.

The formation of amyloid deposits occurs decades before the clinical onset of dementia. Once initiated, the processes associated with AD appear to be independent of age. Monitoring the development of AD processes in ADAD over a period of four decades showed that amyloid deposition can be detected in all persons with mutations in any gene that cause alterations in Aβ processing (*APP*, *PSEN1*, or *PSEN2*) even before the onset of AD symptoms but not in noncarriers of the mutation. Although the clinical and pathological phenotypes of ADAD and LOAD are similar, whether the timing and magnitude of the brain changes observed in ADAD are the same for LOAD remains to be determined [29].

A prospective cohort study of Aβ deposition, neurodegeneration, and cognitive decline in sporadic AD confirmed that Aβ accumulation in the brain (resulting in hippocampal and gray matter volume reduction and memory impairment) is slow with a likely duration of more than two decades [134]. A longitudinal study of cognitive functions, biomarkers in CSF, PET imaging biomarkers (^11^C-PiB, ^18^F-FDG), and MRI confirmed that PiB-PET is associated with AD, neuritic plaques, and diffuse plaques; FDG-PET and MRI have a modest correlation with neuropathologic measures in the AD spectrum [144].

PET measurements show that Aβ accumulates, at least in some AD patients, before abnormal tau accumulation, i.e., an increase in Aβ accumulation may be associated with a consequent increase in tau accumulation [17,95,145]. This indicates that the toxicity of soluble Aβ oligomers may be primarily responsible for the onset of the disease, while soluble tau oligomers may be responsible for the progression of neurodegeneration.

In the development of tau pathophysiology, levels of soluble tau species increase first. Cognitive decline and AD progression are most strongly associated with tau change and phosphorylated tau is sufficient to induce neurodegeneration [146]. Together with the initial development of amyloid pathology, T-tau, P-tau217 and P-tau181 first increase; as neuronal dysfunction increases, P-tau205 and T-tau increase. With the onset of neurodegeneration and cognitive decline, T-tau continues to increase (P-tau217 and P-tau181 decrease) and NFTs arise [147]. Thus, tau pathology may begin together with amyloid pathology long before the onset of clinical symptoms and is associated with an increase in soluble T-tau. Soluble forms of Aβ and tau colocalize early in AD and are associated with disease progression and cognitive decline [148]. However, the model, which considers the accumulation of Aβ and P-tau as primary events in the pathophysiological cascade, cannot be considered confirmed and generally valid because mitochondrial, metabolic, neuroinflammatory, cytoskeletal, and other age-related neuronal alterations may play an even earlier role than Aβ in AD pathogenesis [149,150].

The model of dynamic AD biomarkers [26,142] captures the biomarker magnitude depending on the clinical AD stage and is based on the longitudinal measurement of (i) amyloid pathology (Aβ_42_ in CSF and PET Aβ imaging), (ii) tau pathology (P-tau and T-tau in CSF), (iii) synaptic dysfunction (FDG-PET and fMRI), (iv) brain structure and neurodegeneration (volumetric MRI), and (v) clinical parameters (cognitive decline and dementia). The model assumes that carriers of the *APOE* ε4 allele may have an earlier onset of the Aβ-initiated cascade of pathophysiological processes. Attention is paid to the age-related development of mitochondrial dysfunction. Thus, threshold crossing of mitochondrial dysfunction, Aβ neurotoxicity, and/or tau neurotoxicity appear to be the best candidates for possible primary triggers of AD onset (Figure 2).

## 3. Hypotheses of Alzheimer’s Disease

Progressive neurodegeneration in AD (the loss of synapses and neurons and damage to certain neural circuits in the brain) leads to cognitive dysfunction and dementia. Common mechanisms involved in neurodegeneration in AD include proteinopathy, with Aβ and tau as the major aggregation proteins. Neurodegeneration in neurodegenerative diseases is commonly affected by (1) environmental factors, such as age, diet, exercise, lifestyle, and cognitive reserve; (2) metabolic and oxidative stress; (3) genetics and epigenetics; (4) cerebrovascular dysfunction and blood–brain barrier disorder; (5) neurotoxicity; and (6) neuroinflammation associated with activation of glial cells and/or infiltration of peripheral immune cells and cytokines [151]. Several AD hypotheses have been formulated with focus on various molecular mechanisms underlying the pathophysiology of the disease. Currently, the most attention is paid to the amyloid, tau, neurotransmitter, and mitochondrial cascade hypotheses. Neurovascular, exercise, inflammatory, diabetes, viral, and other AD hypotheses have also been discussed in clinical trials [152,153].

### 3.1. Amyloid Hypothesis

The original amyloid cascade hypothesis postulates that extracellular Aβ deposits represent the fundamental impetus of the AD pathological cascade. The hypothesis supposes that Aβ accumulates in senile plaques, which leads to the formation of neurofibrillary tangles of tau protein, causing brain cell loss and dementia [154,155]. Increased Aβ production and/or impaired Aβ clearance are thought to be the initial triggers of AD pathophysiology in a large part of familial EOAD and in part in sporadic LOAD.

Soluble Aβ oligomers have recently been shown to be responsible for amyloid toxicity in AD [156,157,158] and to disturb neuroplasticity (both synaptic and nonsynaptic). According to the toxic oligomer hypothesis of AD, the cascade of the neurotoxic effects of Aβ and tau includes neuroinflammation, mitochondrial dysfunction, metabolic dysregulation, and protein degradation deficiency, leading to metabolic and oxidative stress, excitotoxicity, defective neurogenesis and neurodevelopment, synaptic damage, axonal transport dysfunction, iron accumulation, and loss of neurons due to apoptosis, necroptosis, necrosis, pyroptosis, ferroptosis, and others [11,159,160,161,162,163] (Figure 3).

The adaptive response hypothesis of AD [164] explains the delayed onset of the disease because of the gradual accumulation and neurotoxic effect of Aβ. According to this hypothesis, Aβ accumulation occurs as a result of an adaptive response to chronic stress. A stress factor can be metabolic dysregulation, especially disturbed metabolism of cholesterol or acetylcholine; insulin resistance; glucose intolerance; obesity; or diabetes. According to this hypothesis, drug treatment should address mitochondrial dysfunction, neuroinflammation, and acetylcholine metabolism. The amyloid cascade hypothesis has been extended to include a complex cellular phase involving feedbacks among astrocytes, microglia, and the vasculature [165].

### 3.2. Amyloid Beta Pathology

APP and its cleaved products are intrinsically linked to cellular dysfunctions associated with AD; reduction of APP and its toxic metabolites could slow the progression of AD. Aβ is produced from APP via cleavage catalyzed by β-secretase 1 (BACE1) and γ-secretase [166,167]; α- and β-secretase compete for APP processing. Aβ may regulate its own production through negative feedback effect on the cleavage of APP by γ-secretase [168]. It is assumed that decreased Aβ clearance and increased occurrence of Aβ oligomers and plaques in the brain are responsible for the Aβ neurotoxicity in AD [9]. Aβ_40_ and Aβ_42_ are the dominant metabolites of APP catalyzed by β- and γ-secretase. Aβ_42_ is more pathogenic. The interaction between Aβ_40_ and Aβ_42_ affects the structural conversion in the process of aggregation. Soluble Aβ_42_ oligomers are stabilized in the presence of Aβ_40_; an equimolar Aβ_42_/Aβ_40_ mixture allows the formation of spherical oligomers, which are more stable and the most toxic Aβ oligomers as evidenced by neurite degeneration and neuronal toxicity [169].

Significant steps in Aβ brain accumulation and oligomerization leading to neurotoxicity and the possibilities for therapeutic intervention are summarized in Figure 4. The Aβ aggregation pathway involves the formation of monomers, oligomers, protofibrils or paranucleus, and mature fibrils, with fibrillar deposits being referred to as amyloid plaques [170,171]. The neurotoxicity of Aβ oligomers may be mediated by (1) their interactions with receptors followed by changes in Aβ clearance or in the activation of various synaptic signaling pathways [172]; (2) binding of Aβ oligomers to the membrane, membrane damage and/or pore formation [173,174]; and (3) intracellular interactions causing mitochondrial dysfunction followed by increased oxidative stress and disruption of other cellular processes [175]. Soluble Aβ oligomers extracted from AD brains have been found to severely disrupt the structure and function of synapses [176]. In transgenic mice, it has been found that Aβ dimers in the absence of plaque pathology can impair the learning and synaptic plasticity that accompany AD [177]. In contrast, synthetic Aβ_42_ monomers promote the survival of developing neurons [156]. Both intracellular and extracellular chaperones can inhibit pathogenic processes associated with Aβ oligomers by binding to them, thereby preventing fibril formation and allowing oligomer sequestration [178].

Several receptors that bind monomeric, oligomeric or fibrillar forms of Aβ have been identified, including the endothelial receptors RAGE (receptor for advanced glycation end products) and LRP1 (low density lipoprotein receptor-related protein 1), which are involved in the clearance of Aβ monomers across the blood–brain barrier, and a number of postsynaptic receptors for Aβ oligomers (e.g., insulin receptor, NGF receptor, cellular prion protein, glutamate NMDA and AMPA receptors, glutamate metabotropic receptor 5, nicotinic acetylcholine receptor, ephrin type-A receptor 4, ephrin type-B receptor 2, and Frizzled receptor in canonical Wnt signaling pathway) [14,165,172,179].

RAGE, which is overexpressed in the brains of AD patients, is involved in Aβ intraneuronal transport (mediates the influx of circulating Aβ) and Aβ-induced mitochondrial dysfunction and neuronal damage [180,181]. Glycated Aβ, which results from increased insulin resistance in AD, appears to be a more suitable ligand for RAGE than unmodified Aβ [182]. Damage associated with RAGE signaling is apparently associated with activation of NADPH oxidases, increased ROS formation, increased expression of cathepsin B (which may function as a β-secretase 1), and asparagine endopeptidase (which cleaves tau protein and APP) [183]. Thus, the RAGE signaling pathway has potential importance in the pathophysiology of AD and may be a target for new AD drugs [184].

There is an association between ApoE4 and Aβ pathology [185]. Neurite outgrowth is stimulated by ApoE-containing lipoprotein particles through binding to LRP1. LRP1 also binds APP and Aβ, affecting their metabolism and thus contributing to the pathogenesis of AD [186]. LRP1 also regulates brain homeostasis and participates in regulating the pathogenic role of ApoE4. Enhancing LRP1-mediated Aβ clearance pathways is a promising therapeutic strategy for AD [187,188].

In summary, many different oligomers with different neurotoxicities have been identified. The main components of Aβ oligomers and Aβ plaques are Aβ_40_ and Aβ_42_, with Aβ_42_ showing much faster self-aggregation into fibrils. Brain cell damage caused by Aβ oligomers encompasses all major aspects of AD neuropathology [189], including oxidative stress, disruption of Ca^2+^ homeostasis, mitochondrial dysfunction, disturbed brain plasticity, aberrant tau phosphorylation, selective neuron death, neuroinflammation, inhibition of axonal transport, receptor redistribution, and insulin resistance. Therapeutic targets may be Aβ oligomers themselves, their receptors and signaling pathways, or downstream effectors, such as tau [179].

### 3.3. Tau Hypothesis

The tau hypothesis is based on the finding that NFTs are composed of phosphorylated tau [190]. Microtubule-associated tau protein stabilizes microtubules, and when hyperphosphorylated, dissociates from microtubules, thereby destabilizing them and disrupting their function. Hyperphosphorylated tau gradually forms dimers, oligomers, protomers, paired helical filaments, and NFTs [191]. According to the tau hypothesis, tau tangles precedes Aβ formation, and the primary cause (initiating factor) of neurodegeneration in AD is microtubule destabilization and the neurotoxic effects of P-tau and its aggregates [192,193]. Tau phosphorylation is controlled by protein kinase and phosphatase activity.

The oligomeric tau hypothesis [194,195] assumes that the main synaptotoxic form of tau is not NFTs, but tau oligomers that form before NFTs and cause neurodegeneration and memory deficits already in the early stages of AD [196,197] (Figure 3 and Figure 5). Tau oligomers appear to serve as templates for the misfolding of native tau and can also alter membrane permeability, thereby causing ion imbalance and can be released into the extracellular space [198,199].

The link between the amyloid and tau hypotheses may be the activation of GSK-3 by overexpression of the enzyme or by the effect of Aβ oligomers. GSK-3 phosphorylates tau, and GSK-3 activity contributes to Aβ production and Aβ-mediated neuronal death [200]. According to the GSK-3 hypothesis of AD, excessive GSK-3 activity is responsible for memory impairment, tau hyperphosphorylation, increased Aβ production, and microglia-mediated neuroinflammatory responses [201].

### 3.4. Tau Protein Pathology

Tau protein is a microtubule-associated protein (MAP) that plays a key role in microtubule stabilization. In neurons, this means that tau promotes axon and dendrite growth and axonal transport. Six tau isoforms are found in the adult brain (3RON, 3R1N, 3R2N, 4RON, 4R1N, and 4R2N). Abnormal tau hyperphosphorylation and its posttranslational modifications lead to neurodegenerative tauopathies. Impaired microtubule function means impairs the development of cellular processes, cell polarity, and intracellular transport, which leads to impaired neuroplasticity and neurodegeneration.

Protein kinases that phosphorylate tau include GSK-3, cyclin-dependent kinase 5 (Cdk5), c-Jun *N*-terminal kinases, casein kinase 1, dual specificity tyrosine-phosphorylation-regulated kinase 1A (Dyrk1A), AMP-activated protein kinase, MAP/microtubule affinity-regulating kinase 4, protein kinase A, and tau-protein kinases. Various protein phosphatases (e.g., PP1, PP2A, PP2B, PP2C, and PP5) may be involved in tau dephosphorylation [202,203,204,205].

In AD brains, tau accumulates in a hyperphosphorylated state [3]. Phosphorylation of tau at multiple sites leads to removal of P-tau from microtubules, destabilization of microtubules, and disruption of a number of cellular processes. Detached tau fragments from microtubules aggregate, which leads to the formation of NFTs. At the same time, P-tau and tau fragments misfold to form tau dimers, trimers, soluble and insoluble oligomers, and paired helical filaments, which eventually form NFTs or toxic tau seeds. Impaired axonal transport and the neurotoxicity of tau oligomers and NFTs leads to loss of neuronal function, apoptosis, and neurodegeneration in AD. The simplified steps leading to the formation of the pathological forms of tau protein in AD are shown in Figure 5.

The mechanism involved in the transcellular spread of tau in neurodegeneration is unclear. Extracellular tau can be transported by exosomes or ectosomes. Tau can be transported into cells by heparin sulfate proteoglycans, by binding to membrane receptors, or by endocytosis. Tau oligomers can also spread between cells via nanotubes [206].

The most significant protein interaction that enhances tau pathology is the interaction with Aβ. Aβ facilitates the phosphorylation, aggregation, mislocalization and accumulation of tau. Mechanisms by which Aβ may enhance tau pathology, leading to neurodegeneration, include activation of specific protein kinases (e.g., GSK-3, Cdk5, and Fyn kinase), neuroinflammation-modulated tau phosphorylation, inhibition of tau degradation by the proteasome, and interactions affecting axonal transport [207]. Aβ and tau appear to act in a synergistic manner, and thus, their effect on neurodegeneration and AD progression is not solely due to additive effects of Aβ and tau [193,208]. The connection between the action of Aβ oligomers and tau hyperphosphorylation can be realized, for example, through the interaction of Aβ oligomers with the complex receptor cellular prion protein/mGluR5, which leads to the activation of Fyn kinase [209]. In addition to Aβ, α-synuclein, some heat shock proteins, immunophilins, cytoplasmic adapter proteins, and peptidyl-prolyl isomerase (Pin1), which regulates Cdk5 dephosphorylation of phosphorylated tau, also regulate tau pathology [202]. Targeting P-tau toxicity and protein interactions that influence tau pathology may be an effective strategy in the development of new AD drugs. For example, a therapeutic target may be the c-Jun *N*-terminal kinase (JNK) signaling pathway, which phosphorylates APP and tau, and whose sustained activation leads to neurodegeneration [210].

### 3.5. Mitochondrial Hypothesis

The mitochondrial cascade hypothesis of sporadic AD [211,212] assumes that mitochondrial dysfunction is a trigger of AD development, i.e., impaired mitochondrial function is a cause of AD, rather than a consequence of it. The mitochondrial hypothesis of AD is based on the key role of mitochondria in the processes of neurodegeneration, including processes regulated by Aβ and tau oligomers. Synaptic mitochondrial dysfunction may be a predisposing factor for AD [213].

Mitochondria-associated ER membranes (MAMs) play a key role in maintaining Ca^2+^ homeostasis, phospholipid and cholesterol metabolism, the import of lipids from ER to mitochondria, initiation of autophagy, mitochondrial division, and apoptosis. MAMs are sites of close reversible membrane contact between mitochondria and the ER and also play a role in the development of neurodegenerative diseases and glucose homeostasis [214,215]. APP processing by presenilin 1 and 2 and γ-secretase occurs predominantly in the MAM [216,217]. Based on this knowledge, the MAM hypothesis was formulated, according to which AD is primarily a communication/connection disorder between the ER and mitochondria [218]. Mitochondrial dysfunction associated with changes in MAM processes involving APP, Aβ, ApoE4, P-tau, mitofusins (localized in the outer mitochondrial membrane), inositol 1,4,5-trisphosphate receptors (releasing Ca^2+^ from the ER to the cytosol), and other structural and functional partners of MAMs is apparently involved in the development of AD [219]. Thus, MAMs may be targets for new AD drugs.

However, the primary cause of AD may not only be mitochondrial dysfunction, but also changes in the functions of other upstream factors, such as ApoE4 and GSK-3, inducing both Aβ and tau pathology. Hypotheses related to the mitochondrial cascade hypothesis are based on the role of metabolites of the mitochondrial TCA and OXPHOS systems, which directly regulate cellular bioenergetics but also play a role in the epigenetic regulation and neuron survival [152]. The mitochondrial toxicity of Aβ and tau oligomers as well as Aβ and tau accumulation associated with mitochondrial dysfunction are key links between the amyloid, tau, and mitochondrial hypotheses. It appears that the pathology of AD may have several initiating factors, including phosphorylated tau, Aβ, and mitochondrial dysfunction, which interact with each other and lead to a similar phenotype.

### 3.6. Mitochondrial Dysfunction

Mitochondria play a key role not only in bioenergetics but also in processes associated with neurotoxicity, neuroinflammation, oxidative stress, neuroplasticity, and neurodegeneration. Age is a major risk factor for AD, and mitochondrial dysfunction is a hallmark of aging [220,221]. Mitochondrial dysfunction in AD includes (i) decreased ATP production, respiratory complex activity, membrane potential (Δψ_m_), Ca^2+^ buffering, axonal mitochondrial transport, and mitochondrial dynamics (fusion and fission) and biogenesis; (ii) increased ROS generation, mtDNA damage, and the mitochondrial permeability transition pore (mPTP) activation; and (iii) defects in mitochondria–endoplasmic reticulum (ER) contacts controlling Ca^2+^ and lipid homeostasis [55,222,223,224,225,226,227,228].

Among the processes of neurodegeneration associated with mitochondrial dysfunction in AD, the most important role is inhibition of the activity of the OXPHOS system, which results in impaired bioenergetics, increased oxidative stress, and reduced membrane potential (Figure 6). Reduced ATP production causes changes in ATP-dependent processes, such as insufficient function of Na^+^K^+^-ATPases, which leads to disturbances in ion transmembrane gradients (release of K^+^ into the extracellular space and entry of Na^+^ into cells) and membrane depolarization, reversal of the function of excitatory amino acid transporter 2 (glutamate transporter), glutamate release, and increased activation of both voltage- and ligand-dependent Ca^2+^-channels. Excessive influx of Ca^2+^ and an increase in its concentration in the cytosol further potentiates the excitotoxicity of glutamate and Ca^2+^-activated phospholipases, proteases, and endonucleases, which degrade the membranes, proteins, and nucleic acids necessary for cell integrity [229].

Excessive influx of Ca^2+^ into the intracellular space during glutamate excitotoxicity can activate the intrinsic apoptosis pathway. Calcineurin activation causes translocation of the pro-apoptotic factor Bad to mitochondria and triggers apoptosis by sequestering the anti-apoptotic factors Bcl-2 and Bcl-xL. The release of cytochrome *c* (cyt *c*) from the intermembrane space of mitochondria leads to formation of the apoptosome. The apoptosome cleaves procaspase-9, and caspase-9 then activates effector caspases. Caspases cleave key components of the cytoskeleton and nucleus, leading to apoptotic cell death. Additionally, after release from mitochondria, apoptosis-inducing factor (AIF) is transported to the nucleus and leads to caspase-independent apoptosis.

Excessive entry of Ca^2+^ into the mitochondrial matrix can cause a decrease in the membrane potential on the inner mitochondrial membrane and induce opening of mPTP, which is a key process in the realization of excitotoxic damage to neurons (e.g., due to mitochondrial swelling, apoptosis, or necrosis). Voltage-dependent anion channel (VDAC), adenine nucleotide translocator (ANT), and other mitochondrial proteins, such as peptidyl-prolyl cis-trans isomerase (PPIF; also known as peptidylprolyl isomerase F or cyclophilin D), phosphate carrier, and ATP synthase are involved in the formation and regulation of the mPTP [224,230]. The mPTP is a heteromultimeric channel spanning both the inner and outer mitochondrial membranes that increases the permeability of mitochondrial membranes to molecules smaller than 1500 Daltons. This results in inner mitochondrial membrane depolarization, OXPHOS uncoupling, and ROS overproduction.

Furthermore, mitochondria are the main producers of reactive oxygen and nitrogen species (RONS), which can damage the cell, including the mitochondria themselves, through DNA damage, lipid peroxidation, and protein oxidation, nitration, and nitrosation. RONS damage membranes and other basic cellular components and alter gene expression, which can lead to cell death via necrotic or apoptotic mechanisms. Oxidative damage is a common result of excitotoxicity, apoptosis, and neuroinflammation.

Both Aβ and tau were reported to localize within mitochondria. A common, direct, and synergistic toxicity of Aβ and tau in synaptic mitochondria has been demonstrated [231,232]. The processes leading to neurodegeneration in AD are associated with the synaptotoxicity and neurotoxicity of (i) Aβ plaques and oligomers, (ii) P-tau aggregates and oligomers, (iii) mitochondrial dysfunction, and (iv) impaired proteostasis and axonal transport [233]. The synaptotoxicity and neurotoxicity of Aβ and tau can be caused by their direct interaction with components of signaling pathways or by neuroinflammation. Neurotoxicity associated with mitochondrial dysfunction is based on disruption of bioenergetics and calcium homeostasis, increased oxidative stress, and induction of the intrinsic apoptosis pathway [30,225,229] (Figure 6).

### 3.7. Synaptoplasticity Hypothesis

Neuroplasticity is a key concept involved in all biological hypotheses of neuropsychiatric and neurodegenerative diseases, including AD. The maladaptive synaptoplasticity hypothesis of AD assumes that the main causative event in the development of the disease is a maladaptive synaptoplastic response, leading to the formation of NFTs in an evolutionarily mediated vulnerable population of human postsynaptic neurons, in response to a reduction in presynaptic input [234]. The clinical and neuropathological features of AD have been attempted to be explained by the hypothesis that amyloid and tau pathology is a secondary consequence of excessive maladaptive plasticity arising in response to remote initiators of AD, which are influenced by many genetic and environmental factors [235].

Neural plasticity is the ability of neurons and neural networks to change their responses to external and internal stimuli based on previous exposure to certain stimuli and the resulting consequences of these previous exposures. The term plasticity includes synaptic plasticity as well as nonsynaptic (structural) plasticity. Nonsynaptic plasticity is mediated by changes in neuronal structures such as the soma, axon, or dendrites. Synaptic plasticity is realized by a change in the strength of synapses, which is frequently associated with long-term potentiation (LTP) or long-term depression (LTD) [236]. Neurobiological processes underlying synaptic plasticity include glutamate signaling through *N*-methyl-d-aspartate (NMDA) and α-amino-3-hydroxy-5-methyl-4-isoxazole propionic acid (AMPA) receptors; neurogenesis; changes in the cell body, axons, dendrites, and neuronal membrane (with impacts on neurotransmitter receptor systems and related intracellular signaling pathways); volume transmission; and glial cell support.

The brain maintains the capacity for cortical reorganization in response to external stimuli (plasticity) throughout life, which is controlled by the inherent properties of neurons, synapses, and glia and by neurotrophic factors and neurotransmitter systems. The magnitude and dynamics of neuroplastic changes are determined by genetic, epigenetic, and environmental factors and change with age. Changes in synaptic plasticity in brain structures, such as the hippocampus, amygdala, striatum, and cortex, have a determining role in learning and memory processes, which are cognitive processes are impaired in AD [14,232,237,238]. The history of the concept of neuroplasticity, which relates to the understanding of AD, was previously reviewed by Teter and Ashford [239].

Disruption of Hebbian synaptic plasticity [240] in AD may be due to aberrant metaplasticity [241], which contributes to the inhibition of LTP and enhancement of LTD in AD [242]. Aβ has been hypothesized to disrupt synaptic plasticity when hydrophobic Aβ oligomers bind/accumulate to neuronal or glial membranes and modulate membrane processes. As a result, there is (i) stimulation of glial cells to release inflammatory cytokines, (ii) a decrease in glutamate reuptake and an increase in its extracellular concentration, and (iii) an increase in the activation of glutamate NMDA receptors, which results in the inhibition of LTP and an increase in LTD and other processes leading to neuronal damage [243]. Analysis of early changes in network connectivity and neuroplasticity that occur before cognitive decline may help in finding presymptomatic biomarkers of AD and therapeutics effective in the early stages of the disease [244].

### 3.8. Neuroinflammation

Mechanisms of onset and/or progression of neurodegeneration in AD involve neuroinflammation. In general, the inflammatory response is associated with the restoration of balance between the upregulated immune-inflammatory response system and the compensatory immune-regulatory reflex system. In the initial stages of AD, inflammation may have a beneficial role in disease pathology, as activated microglia and astrocytes may participate in Aβ clearance. However, if the inflammatory process in the brain is long-term and untreated, it can lead to oxidative and nitrosative stress, damage to neuroplasticity and neurochemistry, and neurodegeneration [245].

Microglia and astrocytes perform multiple functions in the brain, including the formation and refinement of synapses and protective and regenerative responses to infection or brain injury [246]. Cytokines and chemokines produced by microglia and astrocytes also promote the entry of circulating T cells and macrophages from the periphery, which help with pathogen removal. Inflammasomes, signaling complexes that activate inflammatory responses, are part of the innate immune system. Various types of inflammasomes are also present in the cytosol of microglia, astrocytes, and neurons, where their activation contributes to the development and progression of neurodegenerative diseases in which inflammation is considered a common pathophysiological mechanism [1,247,248]. Neuroinflammation may involve exosomes that transport Aβ, tau, inflammatory factors, and other pathogenic substances between microglia, astrocytes, and neurons [249,250]. Exosome-inflammasome crosstalk has been documented: inflammasome activation can regulate exosomes release, and exosomes are an upstream regulator for inflammasome activation or inhibition [251].

The main source of proinflammatory cytokines in the brain is microglia. Activated microglia and proinflammatory cytokines play roles in changes in neural architecture and neurogenesis. The role of neuroinflammation in AD is also demonstrated by the finding that genetic variants associated with an increased risk of developing AD include a number of polymorphisms of genes that are expressed by microglia, such as *ABCA7*, *TREM2*, and *CD33* [252].

Neuroinflammation and reactive gliosis in AD were initially thought to be a response to neuronal loss. More recently, a sustained inflammatory response in the brains of AD patients has been associated with both neurodegeneration and the pathological effects of Aβ and tau, as well as the interconnection of these effects [253,254,255]. Acute and chronic inflammatory responses against Aβ and NFT aggregates induce increased production of nitric oxide (NO), reactive oxygen species (ROS), proinflammatory cytokines and prostaglandins, which can lead to apoptotic neuronal death [256]. An activated/proinflammatory state of microglia or astrocytes is associated with increased concentrations of proinflammatory cytokines and reduced Aβ clearance capacity; accumulation of Aβ may then lead to persistent neuroimmune activation. Proinflammatory cytokines are also involved in tau pathology, likely through increased tau phosphorylation by IL-dependent tau kinases such as Cdk5, GSK-3, and mitogen-activated protein kinase (MAPK) [257,258].

Due to the role of neuroinflammation in the pathophysiology of AD, treatment with anti-inflammatory drugs is offered. The most commonly used anti-inflammatory drugs are nonsteroidal anti-inflammatory drugs (NSAIDs), which inhibit cyclooxygenase (COX) enzymes. However, the effectiveness of NSAIDs and paracetamol on the progression of cognitive decline in MCI and AD has not been confirmed. Only diclofenac use was associated with slower cognitive decline in AD, but likely not through COX inhibition [259].

### 3.9. Metabolic Dysregulation

Defective metabolic pathways in AD include reduced glucose transport (reduced level of glucose transporters 1 and 3), disturbances in glycolysis (downregulation of hexokinase and inhibition of the neuroprotective effects of 2-deoxyglucose caused by increased NADH concentration), mitochondrial dysfunction (decreased level of acetyl-CoA and impaired function of the tricarboxylic acid cycle (TCA) and the OXPHOS system, leading to decreased ATP production and increased ROS production) [21,50]. The main factor regulating glucose transport is insulin and its receptors. Both mitochondrial damage and decreased glucose metabolism can lead to neurodegeneration and cognitive impairment.

Insulin resistance may play an important role in the pathophysiology of AD, as type 2 diabetes mellitus is among the most important risk factors for AD [40] and a significant proportion of AD patients have abnormal serum glucose levels. Insulin resistance increases neuroinflammation and ROS production and downregulates insulin degrading enzyme, which may increase the formation and deposition of both Aβ and P-tau in AD [22,260]. Mitochondrial dysfunction may regulate insulin sensitivity [261]. Insulin resistance may not be the primary cause of AD but may increase the risk of developing and/or worsen disease progression. Aβ and tau pathology and the degree of insulin resistance are regulated by cytokines such as TNF. TNF inhibition may therefore be a suitable therapeutic target for new AD drugs [262].

Of the genetically determined risk factors for LOAD, the function of ApoE4 is the most thoroughly investigated. ApoE functions in the brain include cholesterol transport and Aβ efflux. The ApoE4 isoform is associated with Aβ and NFT accumulation, neuroinflammation, oxidative stress, impaired insulin signaling in the brain [73] and both Aβ- and tau-mediated neurodegeneration [37,74,263,264]. ApoE is expressed in the brain primarily by astrocytes, but neurons express enzymes leading to ApoE fragmentation. Thus, ApoE4 can be cleaved in neurons to generate neurotoxic fragments [265]. The lipid- and receptor-binding regions in ApoE4 fragments cause mitochondrial dysfunction and neurotoxicity, which may be important in metabolic dysregulation in AD [266,267]. The effects of ApoE4 on neuronal insulin signaling, where it interacts with the insulin receptor and disrupts its function, transport, and related signaling, also contribute to the concept of AD as a metabolic disease [73].

### 3.10. Protein Degradation Deficiency

The elimination of inactive or toxic proteins is essential for neuronal viability. A decrease in the efficiency of clearance systems has been described with age and in neurodegenerative diseases such as AD. Protein homeostasis (proteostasis) is maintained by the ubiquitin–proteasome system (UPS) and the autophagy–lysosome system. The UPS system is responsible for the proteolytic degradation of most cellular proteins, including misfolded or damaged proteins. The lysosomal system plays an important role in the degradation of extracellular and intracellular macromolecules. Aβ can be degraded by proteolysis (e.g., by neprilysin, insulin-degrading enzyme, endothelin-converting enzyme, and angiotensin-converting enzyme) or cleared by efflux from the brain to the periphery. The main regulator of Aβ in neurons is the UPS system, either by reducing Aβ production or promoting its proteolytic degradation [268,269]. Disruption of the UPS in AD can lead to abnormal accumulation of Aβ, and at the same time Aβ inhibits proteasomal activity. Thus, the relationship between Aβ and UPS is bidirectional [270]. Defective Aβ- and tau-induced autophagy and mitophagy has been found in AD [271], and Beclin 1 (a key protein in autophagy) is decreased in the early stage of AD [272]. Both the autophagy–lysosome system and the UPS are involved in tau clearance mechanisms [273]. Thus, the UPS system and autophagy are potential therapeutic targets for AD.

## 4. Linking the Amyloid, Tau, and the Mitochondrial Hypotheses of Alzheimer’s Disease

Alzheimer’s amyloidopathy and tauopathy are associated with neurodegeneration through processes such as impairment of energy metabolism, oxidative stress, cytosolic calcium imbalance, apoptosis, and neuroinflammation, which are strongly regulated by mitochondria [274,275]. The interconnection of the amyloid, tau, and mitochondrial hypotheses is evidenced by Aβ-induced or P-tau-induced impairment of mitochondrial function, especially the activity of the respiratory chain.

### 4.1. Integrative Models

A model describing the development of AD biomarkers in relation to each other and to the onset and progression of clinical symptoms has been widely discussed and modified [92,95,158,276]. Acceptance of the fact that Aβ and tau pathology may be initiated independently in LOAD is significant. The dual-pathway hypothesis suggests that Aβ elevation and tau hyperphosphorylation may be independent pathophysiological processes with pathogenic synergy [277]. The Aβ-tau interaction hypothesis proposes that the intrinsic interaction of Aβ and tau (Aβ_42_ increases tau phosphorylation, truncation, and aggregation, while tau can increase Aβ production) determines Aβ and tau pathology, causing synaptic dysfunction, neuronal loss, and cognitive impairment [278]. Impaired neuroplasticity [235] and mitochondrial dysfunction [211] could be a common upstream etiology of AD an are responsible for the effects of multiple risk factors and biomarkers, including Aβ elevation and tau hyperphosphorylation. Most biomarker measurements confirm primary abnormalities in Aβ long before the onset of clinical symptoms of AD. However, in some cases biomarkers preceding Aβ biomarkers may be detected, e.g., hypometabolism measured by FDG PET associated with the nonamyloid effects of ApoE4 [279,280]. Hypometabolism may underlie the mitochondrial hypothesis, but also the cholinergic hypothesis, the metal ion hypothesis, and the neurovascular hypothesis [152]. There are likely multiple mechanisms determining the initiation of Aβ abnormalities. Greater synaptic activity generally leads to greater amyloid deposition, but this is modulated by cognitive reserve and the effects of ApoE4 [281].

The amyloid cascade hypothesis posits that Aβ deposition in the brain drives tau phosphorylation, tangle formation, synapse loss, neuronal death, and the clinical manifestations of AD. The tau cascade hypothesis postulates that tau pathology may be the primary process in the development of AD. The mitochondrial cascade hypothesis posits mitochondrial dysfunction and mitochondria induced Aβ and tau pathology as the primary cause of neurodegeneration in AD. Understanding the link between Aβ and tau pathology and mitochondrial dysfunction will allow the discovery of new drug targets for Alzheimer’s disease with the potential to develop causal drugs. The synergistic/feedback effect of Aβ pathology, mitochondrial dysfunction, and tau pathology leading to neurodegeneration and impairment of brain functions via neurotoxicity and neuroinflammation processes can be considered responsible for the impairment of brain functions in AD.

According to this trial pathway amyloid-tau-mitochondrial integrative hypothesis, it can be assumed that the interaction of risk factors and biomarkers and their mutual potentiation are more decisive for the development of AD than a primary role exerted by any of them individually. This suggests that different AD sufferers may have different initiators of disease development (such as Aβ oligomers, tau oligomers, and mitochondrial dysfunction) that interact and potentiate each other, ultimately contributing to neurodegeneration leading to AD dementia (Figure 7). To confirm this hypothesis, an advanced *in vivo* test of mitochondrial dysfunction in AD patients should be validated and used in longitudinal studies of AD biomarkers.

### 4.2. Mitochondrial Targets of Amyloid Beta

Mitochondrial dysfunction affects Aβ production or clearance. Thus, mitochondrial dysfunction may contribute to amyloidogenesis and trigger neurodegeneration [282], supporting a role for age-dependent mitochondrial dysfunction in AD pathogenesis. Conversely, Aβ can be transported into mitochondria and induce mitochondrial dysfunction [283].

Intracellular Aβ has been confirmed immunohistochemically in organelles, such as the ER, the mitochondria, Nissl bodies, and lipofuscin [284]. Transport of Aβ to the mitochondria [285] allows Aβ to exert direct toxic effects on mitochondrial functions. Mitochondria-controlled cellular processes leading to neuronal damage and death induced by Aβ oligomers include apoptosis, oxidative stress, ion disturbances, neuroinflammation, bioenergetics, and transport.

Oligomeric Aβ can permeabilize membranes; however, a specific uptake is more applied of Aβ into mitochondria can be achieved through translocase of the outer membrane 40 (TOM40) and translocase of the inner membrane 23 (TIM23) [286]. Membrane damage may contribute to the mitochondrial toxicity of Aβ oligomers, e.g., through disruption of calcium homeostasis [287], but specific Aβ interactions with mitochondrial proteins appear to be decisive.

Aβ may induce mitochondrial dysfunction in AD in many ways (Figure 8): (i) impairment of membranes and inhibition of mitochondrial enzymes, such as respiratory complexes, leading to decreased ATP production and increased ROS generation; inhibition of oxoglutarate dehydrogenase complex (OGDC, or α-ketoglutarate dehydrogenase complex), leading to changes in TCA; and interaction with 17β-hydroxysteroid dehydrogenase type 10 (HSD10, formerly known as Aβ binding alcohol dehydrogenase or ABAD), leading to enhancement of Aβ-induced cell stress [288,289]; (ii) increased expression of mitochondrial PPIF, leading to formation of PPIF-mPTP [224,290]; (iii) impairment of mitochondrial Ca^2+^ buffering capacity, leading to calcium overload in the mitochondria, calcium-induced mPTP opening, inhibition of ATP production, increased ROS production, increased cytochrome *c* release, apoptosis, and neuronal injury; (iv) inhibition of preprotein import into mitochondria via the TIM/TOM complex [283,291]; (v) impairment of fusion/fission processes, mitophagy, and mitochondrial movement [292]; and (vi) decreased expression of peroxisome proliferator-activated receptor gamma coactivator 1-alpha (PGC-1α), which is a regulator of mitochondrial biogenesis [275,293,294].

The following have long been demonstrated: (i) progressive accumulation of Aβ in mitochondria [295]; (ii) interaction of Aβ with certain mitochondrial enzymes [296]; and (iii) effect of Aβ on OXPHOS activity leading to ATP depletion, increased ROS production, exacerbation of calcium-dependent formation of the mPTP, and apoptosis [297,298,299,300].

Mitochondrial targets of Aβ include presequence protease (PreP), a metalloprotease responsible for the degradation of Aβ in the matrix, and other Aβ-degrading enzymes [301,302]. Both ANT1 and VDAC1 interact with Aβ and P-tau, which block mitochondrial pores and induce mitochondrial dysfunction in AD [303,304]. Mitochondrial dysfunction and neurodegeneration induced by Aβ or P-tau can be reduced by activation of PGC-1α by a PPARγ agonist, such as metformin, resveratrol, and hopeahainol A (also acting as an acetylcholinesterase inhibitor) [294].

### 4.3. Mitochondrial Targets of Tau

The mitochondrial toxicity of tau oligomers may be mediated by their direct interactions with proteins and by binding of tau oligomers to the mitochondrial membrane and subsequent regulation of membrane proteins causing mitochondrial dysfunction. It has been confirmed that extracellularly applied tau oligomers are internalized by cells [305] and that part of the tau is located on the outer mitochondrial membrane and in the inner mitochondrial space [306], suggesting a potential tau-dependent regulation of mitochondrial functions [307].

Some mitochondrial effects of APP and P-tau are summarized in Figure 8. Interactions of both Aβ and P-tau with mitochondria cause enhanced nitrosylation of dynamin-like protein-1 (Drp1) in AD, leading to impaired mitochondrial dynamics [308,309]. The contribution of soluble tau oligomers to neurodegeneration is thought to lie in their toxicity to neuronal and synaptic function, not only through disruption of mitochondrial transport and dynamics [197] but also through cardiolipin-dependent disruption of mitochondrial membranes [310], and direct interactions with mitochondrial proteins, leading to mitochondrial dysfunction. Overexpression and abnormal mitochondrial accumulation of tau have been observed [311].

The interconnection of the tau and mitochondrial hypothesis is evidenced by the fact that tau pathology causes impairment of mitochondrial transport, affects regulation of mitochondrial dynamics, and significantly affects mitochondrial bioenergetics in AD. Phosphorylated tau reduces complex I and V expression, complex I activity, ATP production, calcium buffering capacity, and Δψ_m_; increases oxidative stress; and affects mPTP formation [25,312].

Aβ-tau-mitochondria interactions lead to uncoupling of oxidative phosphorylation, decreased NAD^+^ reduction and mitochondrial respiration, impaired Δψ_m_, increased ROS production, lower ATP synthesis, changes in mitochondrial dynamics (fission and fusion), malondialdehyde production, opening of the mPTP and swelling of mitochondria, impairment of calcium storage in mitochondria, release of apoptotic factors (such as cytochrome *c* or apoptosis-inducing factor), mitophagy defects, ER stress, decreased mitochondrial biogenesis, and mtDNA oxidation and mutations [222,225,229,313,314,315,316].

## 5. Alzheimer’s Drugs

In neuropsychiatric diseases, when the possibility of biochemical analyzes of the brain is limited, an understanding of the mechanism of drug action is a valuable source of knowledge to better understand the etiology of the disease. Unfortunately, in the case of AD, effective drugs capable of preventing or halting disease progression are not yet available. For effective therapy, it is apparently necessary to start a therapeutic intervention at an early stage of the disease and to use a multifactorial approach that considers the individual initiators of the development of AD. As of January 2022, there were 143 agents in clinical trials for AD drugs [317]; these are mostly disease-modifying drugs, symptomatic cognitive enhancers, and neuropsychiatric drugs. This chapter is focused on approved AD drugs and selected potential AD drugs; supplements and comorbid medications are also mentioned.

### 5.1. Cellular Drug Targets

The cellular targets of newly developed AD drugs include Aβ pathology (inhibitors of β- or γ-secretase, α-secretase modulators, aggregation inhibitors, metal interfering drugs, Aβ clearance enhancers, and Aβ receptors such as RAGE and LRP1), tau pathology (inhibitors of tau hyperphosphorylation, activators of phosphatases, aggregation inhibitors, tau clearance enhancers, modulators of tau glycosylation, and inhibitors of intercellular transfer and tau uptake), neuroinflammation, oxidative stress, monoamine oxidase type B (MAO-B) and other mitochondrial proteins, glutamate NMDA receptors, nicotinic acetylcholine receptors, metal chelation, and JNK and other signaling pathways [30,164,318,319,320]. The effectiveness of pharmacotherapy can be enhanced by combining it with nondrug interventions (physical and mental activity and diet) and supplements, that may have beneficial effects on brain function due to their broad multitarget effects [321]. Recently, it is hypothesized that drug targeting of the gut microbiota could be effective in alleviating neuroinflammation in AD [322].

In addition to acetylcholine and glutamate systems, which are affected by current AD drugs, the monoamine system has frequently been proposed as a possible target for the treatment of AD [323], e.g., through the modulation of MAO activity [324], serotonin uptake [152], and catecholaminergic function [325].

Mitochondrial proteins and membranes are attractive targets for new AD drugs. Regulation of the following mitochondrial functions by small molecules appears promising in AD treatment: protein acetylation (which is regulated by coenzyme NAD^+^, tricarboxylic acid cycle activity, OXPHOS, sirtuins, and nicotinamide phosphoribosyl transferase), inhibition of the mitochondrial calcium uniporter, inhibition of mPTP opening, and modulation of mitochondrial morphology and dynamics (by the outer mitochondrial membrane fusion protein mitofusin, the inner membrane fusion protein dynamin-like 120 kDa, and the fission protein DRP1) [30,224,326]. Advanced assays are validated for finding and confirming mitochondrial dysfunction in AD [327] and for *in vitro* and *in vivo* testing of the mitochondrial effects of new potential AD drugs [328,329]. The *in vitro* effect of various cognitives, nootropics, and newly synthesized drugs on MAO activity and mitochondrial respiration has been confirmed [330,331].

The insufficient success of antibodies and vaccines directed against Aβ in reducing cognitive decline [332,333] indicates that the treatments are being administered too late and that it is necessary to prevent the early formation of Aβ oligomers [334]. Additionally, during the therapeutic reduction of Aβ levels in the brain it must not be forgotten that endogenous Aβ is necessary in a healthy brain for normal hippocampal synaptic plasticity and memory [335]. Additionally, anti-tau therapies have not been fully successful and are under investigation [205,336,337,338]. Promising therapeutic goal of new causal AD drugs is elimination of both the formation, spreading, and mitochondrial toxicity of Aβ oligomers [339,340,341] and P-tau oligomers [25].

The multitarget-directed ligand (MTDL) approach is frequently used in the development of new AD drugs. Newly tested MTDLs affect various pathways involved in the development of AD, mainly focusing on reducing cholinergic depletion, glutamate toxicity, Aβ aggregation, tau hyperphosphorylation, and oxidative stress [342,343,344,345,346]. For example, ladostigil, which acts as a reversible cholinesterase inhibitor and an irreversible MAO-B inhibitor, has entered clinical trials; it has neuroprotective effects, increases the expression of neurotrophic factors, induces neurogenesis, and has antidepressant effects [347,348].

### 5.2. Approved Drugs

Drugs that are approved for the treatment of AD symptoms are donepezil, rivastigmine, galantamine (cholinesterase inhibitors), memantine (low-affinity uncompetitive antagonist of glutamate NMDA receptors), and combinations of memantine and donepezil (Table 1). While memantine works by reducing glutamate excitotoxicity, other approved AD drugs temporarily improve cognitive symptoms by increasing the availability of acetylcholine in the brain. They can help reduce some of the symptoms of AD and help with certain behavioral problems. Cholinesterase inhibitors exhibit neuroprotective effects by activating the PI3K/Akt/mTOR (phosphoinositide 3-kinase/protein kinase B/mammalian target of rapamycin) neuronal survival pathway, which leads to inhibition of GSK-3 (reduction of tau hyperphosphorylation and formation of NFTs), inhibition of Forkhead box O (FoxO) transcription factors and pro-apoptotic factor BAD (increase of cell survival), and activation of the mTOR complex 1 (inhibition of autophagy) and nuclear factor erythroid 2-related factor 2 (NRF2; increase in antioxidant response) [345].

Aducanumab has recently been conditionally approved by the US Food and Drug Administration (FDA) for the treatment of AD and, thus far, is the only drug that can potentially slow the progression of AD. This is a disease-modifying immunotherapy based on the removal of abnormal Aβ (blocking Aβ aggregation to fibrils) and the reduction of the number of plaques in the brain, which can help slow the progression of AD. The mechanism of action of aducanumab (significant reduction of Aβ in the cerebral cortex) supports the amyloid hypothesis of AD [349]; however, its effect on the progression of cognitive decline or dementia in AD cannot be considered proven [319,350].

**Table 1 biomolecules-12-01676-t001:** Alzheimer’s disease drugs approved for medical use.

Drug	Primary Action Included	Reference
Rivastigmine	Cholinesterase inhibition	[351]
Galantamine		[352]
Donepezil		[352]
Memantine	Blockade of NMDA receptor	[353]
Aducanumab *	Monoclonal antibody directed at brain Aβ plaques and oligomers	[349]

Aβ—amyloid beta; NMDA—*N*-methyl-d-aspartate. * Approved for medical use in the United States by the Food and Drug Administration in 2021.

### 5.3. Drug Candidates

Current symptomatic treatment of AD is focused on the regulation of brain neurochemistry and allows only alleviation of the progression of neurodegeneration. The discovery of new causal AD drugs can be expected, primarily in the field of neurodegeneration regulators related to (i) inhibition of the formation of toxic Aβ oligomers, regulation of APP production and proteolysis, removal of Aβ from the brain, and regulation of signaling pathways activated by Aβ oligomers; (ii) synthesis, phosphorylation, and aggregation of tau; (iii) affecting mitochondrial biogenesis, transport, dynamics, and function; and (iv) regulating the activity of neuronal survival signaling pathways, such as the PI3K/Akt/mTOR pathway and other neuroprotective pathways. Selected AD drug candidates with an indication of the primary biochemical effects associated with potential therapeutic effects (improving of cognition and memory) are shown in Table 2.

The largest group of drugs tested in phase 3 of AD drug development consists of disease-modifying small molecules [105], which include substances modulating (i) metabolism and bioenergetics, e.g., metformin [354,355], semaglutide [356], and tricaprylin [357,358]; (ii) oxidative stress and blood flow in the brain, e.g., omega-3 polyunsaturated fatty acids [359] and ethyl eicosapentaenoate (icosapent ethyl) [360]; (iii) synaptic plasticity and neuroprotection, e.g., blarcamesine [361,362], atuzaginstat [363], AGB101 (levetiracetam) [364,365], and simufilam [366]; (iv) Aβ pathology, e.g., ALZ-801 (valiltramiprosate), an oral prodrug of homotaurine that blocks the formation of toxic Aβ oligomers [334,367,368]; (v) neuroinflammation, e.g., NE3107 [369] and curcumin [337,370,371,372,373]; and more. Disease-modifying drugs are also the monoclonal antibodies directed at Aβ; besides aducanumab [374] it is also gantenerumab [374,375], lecanemab [375], donanemab [376], and solanezumab [374,377], which have shown some efficiency and are tested in phase 3 clinical trials [317].

Other groups of new clinically tested AD drugs are cognitive enhancers and drugs affecting neuropsychiatric symptoms; these substances mostly primarily act on neurotransmitter receptors. Among the tested cognitive enhancers is caffeine [378,379], guanfacine [380], and octohydroaminoacridine succinate [381], but also nicotine [382,383], CST-2032 (β_2_-adrenergic receptor agonist), and AD-35 [384]. Neuropsychiatric drugs tested for use in AD include escitalopram [385], brexpiprazole [386], dextromethorphan/bupropion (AXS-05) [387], and deudextromethorphan/quinidine (d-DXM/Q) [388]; this group also includes drugs acting through the endocannabinoid system, e.g., nabilone, dronabinol, and cannabidiol (CBD) [389] (Table 2).

**Table 2 biomolecules-12-01676-t002:** Selected potential Alzheimer’s disease drugs.

Drug	Primary Brain Action Included	Clinical Trials	Reference
Disease-modifying molecules			
Metformin	Improving of glucose metabolism; mitochondrial complex I inhibition	Phase 3	[354,355]
Semaglutide	Glucagon-like peptide-1 agonism; improving of glycemic control; anti-inflammation	Phase 3	[356]
Tricaprylin	Ketosis and improving of mitochondrial function	Phase 3	[357,358]
Omega-3 PUFA	Anti-inflammation; antioxidant; synaptic plasticity; cerebrovascular function; blood flow	Phase 3	[359]
Icosapent ethyl		Phase 3	[360]
Blarcamesine	σ_1_ receptor agonism; reduction of Aβ and NFTs; anti-inflammation; amelioration of mitochondrial dysfunction and oxidative stress; antiapoptotic; induction of neurogenesis	Phase 3	[361,362]
Atuzaginstat	Inhibition of gingipains; reduction of neurodegeneration and neuroinflammation	Phase 3	[363]
AGB101	Inhibition of SV2A; reduction of Aβ pathology	Phase 3	[364,365]
Simufilam	Reduction of P-tau and Aβ aggregates; reduction of α7 nicotinic acetylcholine, NMDA, and insulin receptor dysfunction	Phase 3	[366]
Homotaurine (ALZ-801)	Inhibition of Aβ aggregation; GABA_A_ receptor agonism	Phase 3	[367,368]
NE3107	Anti-inflammation	Phase 3	[369]
Curcumin	Anti-inflammation; antioxidant; dual inhibition of Aβ and tau aggregation	Phase 2	[337,371,372,373]
Gantenerumab	Monoclonal antibody directed at brain Aβ	Phase 3	[374,375]
Lecanemab		Phase 3	[375]
Donanemab		Phase 3	[376]
Solanezumab		Phase 3	[374,377]
Neuropsychiatric drugs			
Escitalopram	Selective serotonin reuptake inhibition	Phase 3	[385]
Brexpiprazole	D_2_ receptor antagonism	Phase 3	[386]
Dextromethorphan/bupropion	NMDA receptor agonism	Phase 3	[387]
Deudextromethorphan/quinidine	Agonism σ_1_ and antagonism NMDA receptor; serotonin–norepinephrine reuptake inhibition	Phase 3	[388]
Nabilone	Cannabinoid receptors agonism	Phase 3	[389]
Dronabinol		Phase 2	[389]
Cannabidiol		Phase 2	[389,390]
Cognitive enhancers			
Caffeine	Antagonism of adenosine A_2A_ receptor; mitochondrial function	Phase 3	[378,379]
Guanfacine	α_2A_-adrenergic receptor agonism	Phase 3	[380]
Octohydroaminoacridine succinate	Acetylcholinesterase inhibition	Phase 3	[381]
Nicotine	Nicotinic acetylcholine receptor agonism	Phase 2	[382,383]
AD-35	Acetylcholinesterase inhibition; disassembly of Aβ aggregates	Phase 2	[384]

Aβ—amyloid beta; GABA—γ-aminobutyric acid; icosapent ethyl—ethyl eicosapentaenoate; NFT—neurofibrillary tangle; NMDA—*N*-methyl-d-aspartate; P-tau—phosphorylated tau protein; SV2A—synaptic vesicle glycoprotein 2A.

### 5.4. Supplements

People with AD have increased comorbidities [391] and increased medication use for both other neuropsychiatric conditions (such as depression, anxiety, psychosis, and opioid-treated pain) and some metabolic, endocrine, and neurological conditions (such as diabetes, hyperthyroidism, epilepsy, and Parkinson’s disease) [392]. Statins, antihypertensive drugs, antidiabetic agents, cerebrolysin, psychostimulants, and herbal medication seem to be effective in improving cognitive function in AD, but the evidence is limited [393]. A systematic review of the use of supplements did not show a reduction in the risk for cognitive decline [394].

The mechanisms of action of supplements are complex and not well known, but it can be expected that their potential anti-amyloid, anti-tau, neurochemical, mitochondrial, antioxidant, and anti-inflammatory effects may be involved in mitigating the progression of cognitive impairment in AD. Mitochondrial-targeted therapies used in AD models and clinical trials include antioxidants (α-lipoic acid, *N*-acetyl-cysteine, Ginkgo biloba, Szeto-Schiller tetrapeptides 31, catalase, vitamin E and C, selenium, coenzyme Q_10_ and mitoquinone, astaxanthin, apocynin, and SkQ1), nicotinamide adenine dinucleotide (NAD), latrepirdine, 2-deoxyglucose, pioglitazone, oxaloacetate, resveratrol, quercetin, and mitophagy stimulators [222,395]. Mitochondrial function can also be regulated by melatonin [396], methylene blue [397], carotenoids [398], red gingseng [399], and oxaloacetate [400]. Most of these supplements have multiple effects, e.g., inhibition of tau aggregation may be involved in the effects of methylene blue [337,401], red gingseng [402], crocin [403], cinnamaldehyde and epicatechin [404], purpurin [405], and folate [406], and inhibition of Aβ toxicity may be involved in the effects of resveratrol [407], huperzine A (cholinesterase inhibitor and NMDA receptor antagonist) [408,409] and carvacrol (anti-acetylcholinesterase, antioxidant, and neuroprotective properties) [410,411].

In summary, attention is paid both to new disease-modifying drugs and the possibilities of the therapeutic potential of methylene blue, organophosphorus compounds, natural products, and Indian and Chinese medicine [397,412,413,414]. However, their effects on cognitive deficits in AD are still far from proven. In addition to AD pharmaceuticals, neurochemistry and neuroplasticity can also be influenced by lifestyle (exercise), appropriate treatment of comorbid diseases (e.g., antihypertensive and antidiabetic medications reducing cerebrovascular pathology), psychosocial interventions, and diet.

## 6. Conclusions

Aβ and tau pathology and mitochondrial dysfunction are implicated in the etiology of AD. The neurotoxicity of soluble Aβ oligomers is thought to be predominant in the preclinical stage of the disease, while soluble tau oligomers are involved in the progression of neurodegeneration in the prodromal period and in dementia [24,95]. However, it cannot be ruled out, at least in some AD cases, that Aβ toxicity is initiated and potentiated by mitochondrial dysfunction and/or P-tau. The possibility of a causal treatment for AD is based on the recognition and elimination or reduction of the primary causes of the disease, which can be derived from risk factors and biomarkers measurable at an early stage of the disease. Insights into risk factors, biomarkers, the time course of biomarkers, Aβ and tau neurotoxicity, metabolic and mitochondrial dysregulation, and disruption of signaling pathways and neuroplasticity, along with new insights into the mechanisms of neurodegeneration and the mitochondrial targets of Aβ and tau, support an interlinking of the amyloid, tau, and mitochondrial hypotheses. The integrative amyloid-tau-mitochondrial hypothesis posits that risk factors and metabolic changes trigger a primary specific cause of AD development, which is mitochondrial dysfunction, neurotoxicity of Aβ oligomers or neurotoxicity of tau oligomers, and the mutual feedback and synergistic interactions of these primary causes lead to neurodegeneration.

Pharmacological intervention with the potential to be causal includes targeting on the production, elimination, and spread of (i) Aβ oligomers to prevent disease onset [334], (ii) tau oligomers and NFTs to eliminate disease progression [415], and (iii) mitochondrial dysfunction to reduce the progression of neurodegeneration [30]. Based on the integrated amyloid-tau-mitochondrial hypothesis, understanding the mechanism underlying the mitochondrial action of Aβ and tau oligomers and assessing the protective effects of novel AD drugs against the mitochondrial toxicity of Aβ and tau is essential for the successful development of new AD drugs that can effectively/causally treat the disease. It seems appropriate to target signaling pathways involved in feedback links between mitochondria and Aβ and tau; however, these pathways must first be examined in detail.

## Figures and Tables

**Figure 1 biomolecules-12-01676-f001:**
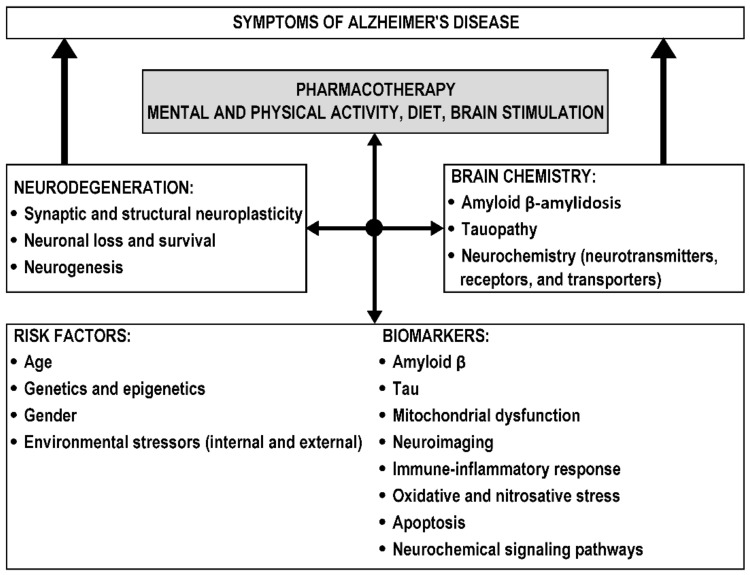
Linking risk factors, biomarkers, and associated brain changes in the development of Alzheimer’s disease (AD). The diagram shows that different risk factors and biomarkers can converge to induce neurodegeneration and impaired brain chemistry, which are responsible for disease symptoms. Double-sided arrows indicate mutual connections and feedback effects of risk factors and biomarkers, neurodegeneration, brain chemistry, and AD therapy.

**Figure 3 biomolecules-12-01676-f003:**
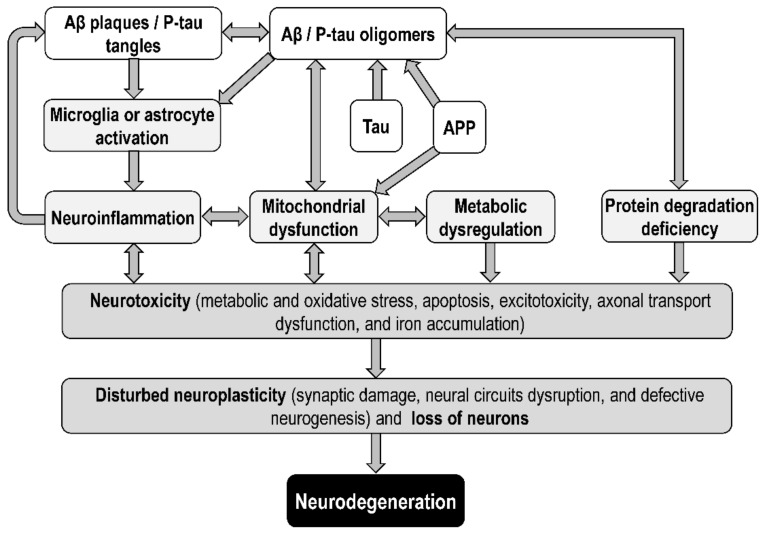
Neurotoxic effects of amyloid precursor protein (APP), amyloid beta (Aβ) and phosphorylated tau (P-tau) leading to neurodegeneration in Alzheimer’s disease.

**Figure 4 biomolecules-12-01676-f004:**
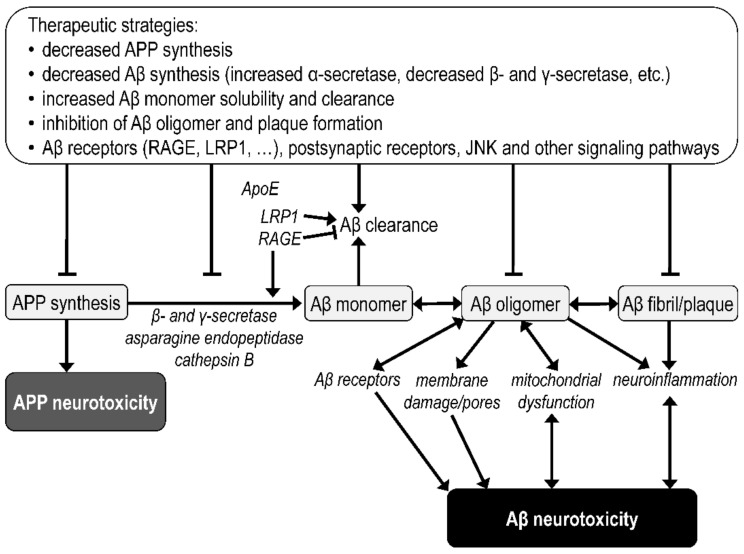
The amyloid beta aggregation pathway, neurotoxicity triggers, and related therapeutic strategies. Aβ—amyloid beta; ApoE—apolipoprotein E; APP—amyloid precursor protein; LRP1—low density lipoprotein receptor-related protein 1; RAGE—receptor for advanced glycation end products.

**Figure 5 biomolecules-12-01676-f005:**
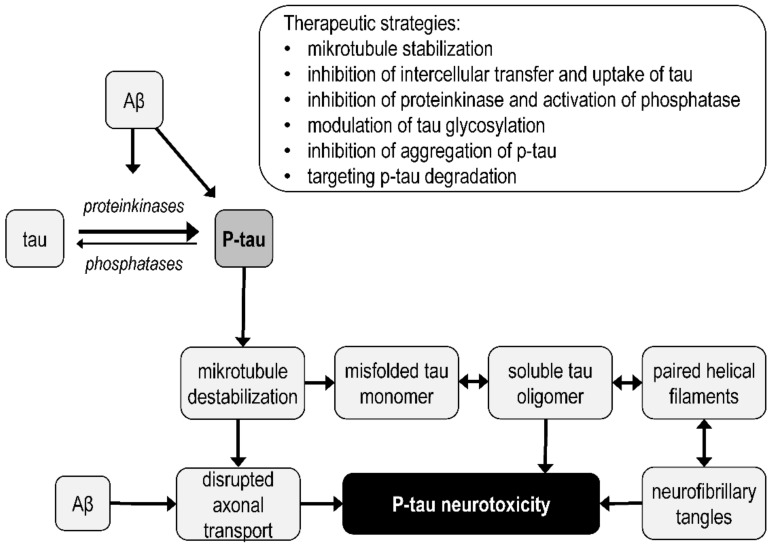
Formation of the toxic forms of tau protein and related therapeutic strategies in Alzheimer’s disease. Aβ—amyloid beta; P-tau—phosphorylated protein tau.

**Figure 6 biomolecules-12-01676-f006:**
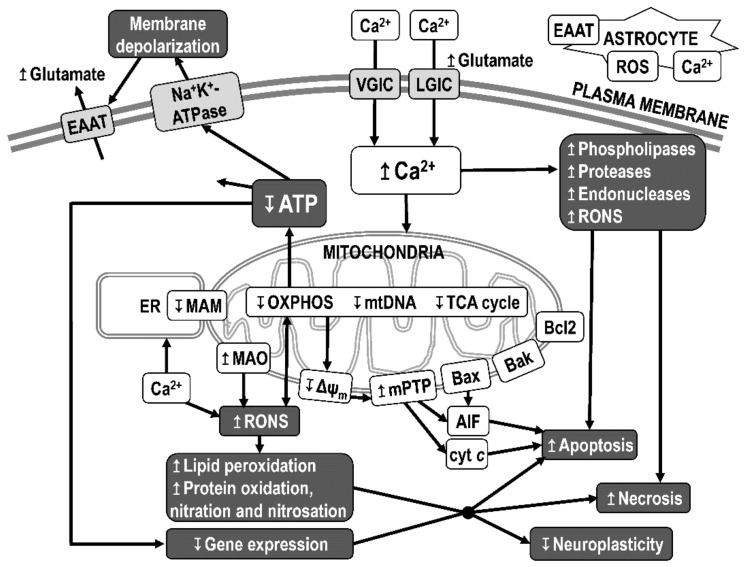
Mitochondrial dysfunction leading to neurodegeneration. The mitochondria-driven processes leading to neuronal damage and cell death are reduced ATP production, increased production of reactive oxygen and nitrogen species (RONS), initiation of apoptotic processes, and disturbed calcium homeostasis. Decreased ATP production leads to disruption of ATP-dependent processes, including restoration and maintenance of membrane potential. Membrane depolarization alters the function of excitatory amino acid transporters and leads to the activation of voltage-gated ion channels (VGICs) and ligand-gated calcium channels (LGICs) and to increased Ca^2+^ entry into the cell. Intracellular calcium acts as a second messenger for the activation of a number of cellular processes and is transported from the cytosol by ion pumps to the extracellular space, to the endoplasmic reticulum (ER), and to the mitochondrial matrix. An excessive concentration of cytosolic free Ca^2+^ can have a neurotoxic effect by increasing the activation of phospholipases, proteases, and endonucleases. Excessive accumulation of Ca^2+^ in the mitochondrial matrix causes a decrease in the membrane potential on the inner mitochondrial membrane (Δψ_m_), a decrease in ATP production, an increase in the production of reactive oxygen species (ROS), the opening of mitochondrial permeability transition pores (mPTPs), and the release of cytochrome *c* (cyt *c*) and other pro-apoptotic factors (triggering caspase-dependent apoptosis) and of apoptosis-inducing factor (AIF) (triggering caspase-independent apoptosis). Mitochondria in the brain are also a target of nitric oxide. Bax, Bak, Bad—Bcl-2 family proapoptotic factors; Bcl-2—Bcl-2 family antiapoptotic factor; EAAT—excitatory amino acid transporter; MAM—mitochondria-associated ER membrane; MAO—monoamine oxidase.

**Figure 7 biomolecules-12-01676-f007:**
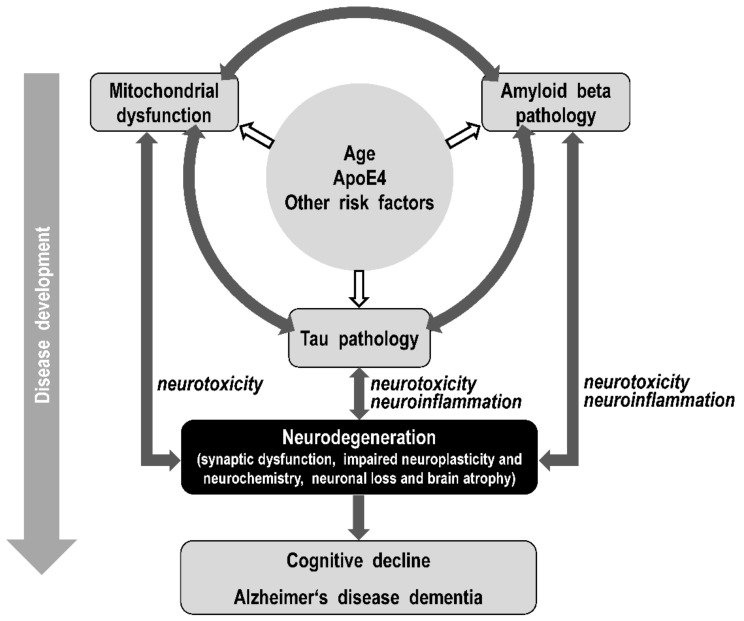
Schematic showing the link between mitochondrial dysfunction and the neurotoxicity of amyloid beta and tau oligomers in the development of Alzheimer’s disease. ApoE—apolipoprotein E.

**Figure 8 biomolecules-12-01676-f008:**
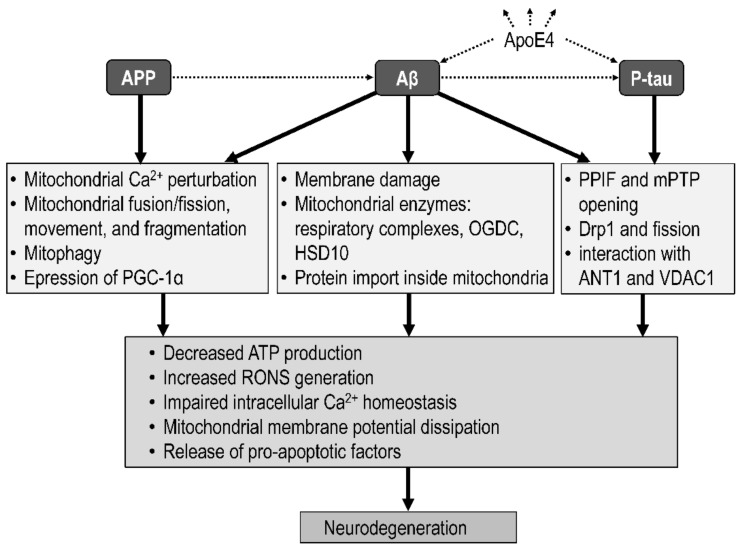
Amyloid beta (Aβ)-induced, amyloid precursor protein (APP)-induced, and tau-induced mitochondrial dysfunction. A direct potentiating effect of ApoE4 on Aβ and tau pathology is indicated. Aβ can cause mitochondrial dysfunction by disrupting the integrity of the cytoplasmic membrane; inhibiting the activity of the oxidative phosphorylation system, HSD10, and OGDC; and inhibiting the import of proteins into mitochondria. Both APP and Aβ disrupt the ability of mitochondria to buffer Ca^2+^, impair mitochondrial dynamics, and decrease the expression of peroxisome PGC-1α. Both Aβ and P-tau interact with ANT1 and VDAC1, increase PPIF expression and mPTP opening and cause increased nitrosylation of Drp1, leading to increased mitochondrial fission and neurodegeneration. ANT1—adenine nucleotide translocator 1; ApoE4—apolipoprotein E4; Drp1—dynamin-like protein-1; HSD10—17β-hydroxysteroid dehydrogenase type 10; mPTP—mitochondrial permeability transition pore OGDC—oxoglutarate dehydrogenase complex; PGC-1α—peroxisome proliferator-activated receptor gamma coactivator 1-alpha; PPIF—peptidyl-prolyl cis-trans isomerase, mitochondrial; RONS—reactive oxygen and nitrogen species; VDAC1—voltage-dependent anion channel 1.

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
