# Peer review of "Linking the Amyloid, Tau, and Mitochondrial Hypotheses of Alzheimer’s Disease and Identifying Promising Drug Targets"

_biomolecules, 2022, doi:10.3390/biom12111676_

Round 1
Reviewer 1 Report
The author has carried out an extensive review of the literature that is reflected in the length of the manuscript and the number of citations (384). The manuscript is fairly well written but needs some editing and deleting some sections to make it easier to read. The length and some repetitive information found along the paper can leave the reader feeling bored.
1. I would suggest to add and index that helps the reader to follow the reading and know what is going to be found. The author needs to add some subheadings since the sections or chapters are too long.
2. The chapter “Risk factors and Biomarkers” is confusing. ApoE4 is not a biomarker as it is indicated in figure 2. Neither are biomarkers “Epigenetics” or “Omics”. It would be more clear to dedicate one chapter to “Risk factors” and a different one to “Biomarkers”
3. The subheading 3.1. (Mechanisms of neurodegeneration) should be eliminated, it is more an introduction of the chapter 3. (Neurodegeneration in Alzheimer’s disease). The last section (3.8. Neuroplasticity) does not belong to this section of Neurodegeneration.
4. The chapter 4 (“Hypothesis”) overlaps with chapter 3, and, in any case, it would make more sense to have first the hypothesis, and later the molecular mechanisms. Or include the hypothesis in the chapter 3. To keep both sections as they are is a bit repetitive.
5. In the chapter 5, please clearly separate the drugs that are approved for clinical use from those whose effectiveness has not been demonstrated. Is not fear to put in the same table the drugs that have been tested in humans and are approved for patients with compounds that have been tested only in animal models, such as red ginseng or gingko biloba.
6. Is not clear to me that the author means with “5.2. Therapeutic strategies”? What is the difference with previous subheading?
7. The chapter 5.3. Promising drug targets based on amyloid-tau-mitochondrial hypothesis” is messy, untidy and again repetitive. 3
Author Response
Point 1: I would suggest to add and index that helps the reader to follow the reading and know what is going to be found. The author needs to add some subheadings since the sections or chapters are too long.
Response 1: An index has been added; one new chapter and several new subheadings were created.
Point 2: The chapter “Risk factors and Biomarkers” is confusing. ApoE4 is not a biomarker as it is indicated in figure 2. Neither are biomarkers “Epigenetics” or “Omics”. It would be more clear to dedicate one chapter to “Risk factors” and a different one to “Biomarkers”
Response 2: ApoE4, “Epigenetics”, and “Omics” were removed from the biomarkers in Figs. 1 and 2. Fig. 2 was removed, and risk factors and biomarkers were separated to different subheadings 2.1. (Risk factors) and 2.2. (Biomarkers). Redundant sentences were deleted from the subheading 2.2. (Biomarkers).
Point 3: The subheading 3.1. (Mechanisms of neurodegeneration) should be eliminated, it is more an introduction of the chapter 3. (Neurodegeneration in Alzheimer’s disease). The last section (3.8. Neuroplasticity) does not belong to this section of Neurodegeneration.
Response 3: The subheading 3.1. (Mechanisms of neurodegeneration) was eliminated; the text of this subheading was shortened and used as an introduction of the chapter 3. (Hypotheses of Alzheimer's disease). The subheading 3.8. (Neuroplasticity) was shortened and used as an explanation for the maladaptive synaptoplasticity hypothesis of AD in the subheading 3.7. (Synaptoplasticity hypothesis) in the chapter 3. (Hypotheses of Alzheimer's disease).
Point 4: The chapter 4 (“Hypothesis”) overlaps with chapter 3, and, in any case, it would make more sense to have first the hypothesis, and later the molecular mechanisms. Or include the hypothesis in the chapter 3. To keep both sections as they are is a bit repetitive.
Response 4: Earlier chapters 3 (Neurodegeneration in Alzheimer's disease) and 4 (Hypotheses) were merged into chapter 3 (Hypotheses of Alzheimer's disease). Redundant text has been removed and the paragraphs have been ordered so that the hypothesis is presented first and the molecular mechanisms later. The oligomeric tau hypothesis was added to the subheading "Tau hypothesis". Former subheading 4.4. (Linking the amyloid, tau, and the mitochondrial hypotheses) was transformed into a separate chapter 4. (Linking the amyloid, tau, and the mitochondrial hypotheses of Alzheimer's disease).
Point 5: In the chapter 5, please clearly separate the drugs that are approved for clinical use from those whose effectiveness has not been demonstrated. Is not fear to put in the same table the drugs that have been tested in humans and are approved for patients with compounds that have been tested only in animal models, such as red ginseng or gingko biloba.
Response 5: Table 1 (Approved drugs and selected potential drugs and adjuvants affecting amyloid beta, tau, and mitochondria.) was divided into two parts: Table 1. (Alzheimer's disease drugs approved for medical use) and Table 2 (Selected potential Alzheimer's disease drugs). Aducanumab has been added to Table 1. All substances that are not clinically evaluated have been removed from Table 2 - only drugs evaluated in phase 3 or 2 of AD drug development are listed in Table 2. A column has been added to Table 2 indicating the phase of clinical trial of the drug.
Point 6: Is not clear to me that the author means with “5.2. Therapeutic strategies”? What is the difference with previous subheading?
Response 6: The subheading 5.2. (Therapeutic strategies) was revised to subheading 5.1. (Cellular drug targets).
Point 7: The chapter 5.3. Promising drug targets based on amyloid-tau-mitochondrial hypothesis” is messy, untidy and again repetitive. 3
Response 7: The entire chapter 5 (Alzheimer's drugs and adjuvants) has been reworked into chapter 5 (Alzheimer's drugs). Redundant information was deleted, the chapter was supplemented and transformed into subheadings: 5.1. (Cellular drug targets), 5.2. (Approved drugs), 5.3. (Drug candidates), and 5.4 (Supplements). Chapter 5 (Alzheimer's drugs) is now focused on approved AD drugs and selected potential AD drugs; supplements and comorbid medications are only briefly mentioned. The chapter 5.3. (Promising drug targets based on amyloid-tau-mitochondrial hypothesis) was canceled because it contained repetitive information and was partly speculative. The message about promising drug targets is now in subheading 5.1. (Cellular drug targets).
Reviewer 2 Report
1. Abstract: the author should include the article search and inclusion criteria, various domains used for searching the articles and number of articles included in writing this review.
2. Page 2, Line 71-72, rewrite the sentence as it has many ‘and’.
3. Figure-1, some examples of specific biomarkers can be included in each category of mitochondrial dysfunction, neuroimaging etc. as in figure-2. Why the author did not mention the brain insulin signalling pathway (insulin sensitivity and glucose uptake) as the Biomarker in figure-1 & 2 and discussed in the review? The link between insulin sensitivity and AD development is evident, and AD is informally named type-3 diabetes mellitus.
4. Page 3, Line 90-105 can be minimized as much as possible because it covers mostly general explanations.
5. Page 4, Page 137-142, the whole paragraph should have a citation of the reference. The author should explain between the less significant and little significant. In both cases, NSAIDs are mentioned as little and less significantly correlated with AD development. Some of the behavioural aspects were also included in the correlation, such as agreeableness, conscientiousness and openness. Secondly, how this information is useful for your review.
6. Mitochondrial dysfunction is well correlated with the onset/progression of neurodegenerative diseases such as AD and Parkinson’s disease (PD). Figure 3 is a hypothetical diagram proposing mitochondrial dysfunction gradually develops throughout the ageing process. The x-axis is unclear as it has not mentioned “years of age" or “birth to death” or any other meaning. Further, can this hypothetical diagram apply to PD?
7. Page-10 to 19, the information on neurodegeneration in AD is well available in other published materials and the content can be minimized with a specific focus on major areas.
8. The review is very long and should focus on the linkage between A-beta-Tau and mitochondrial dysfunction. The author is advised to minimize the content making it more specific to the title. Otherwise, the reader is lost in the middle in understanding other contents.
Author Response
Point 1: Abstract: the author should include the article search and inclusion criteria, various domains used for searching the articles and number of articles included in writing this review.
Response 1: Article search was performed mainly with the help of PubMed with the search according to the listed keywords. Since this is not a systematic review or meta-analysis and due to the limited scope of the Abstract, brief information has only been added to the last paragraph of the Introduction.
Point 2: Page 2, Line 71-72, rewrite the sentence as it has many ‘and’.
Response 2: The sentence has been rewritten.
Point 3: Figure-1, some examples of specific biomarkers can be included in each category of mitochondrial dysfunction, neuroimaging etc. as in figure-2. Why the author did not mention the brain insulin signalling pathway (insulin sensitivity and glucose uptake) as the Biomarker in figure-1 & 2 and discussed in the review? The link between insulin sensitivity and AD development is evident, and AD is informally named type-3 diabetes mellitus.
Response 3:
- Only categories/groups of risk factors and biomarkers are shown in Figure 1, as all risk factors and biomarkers are listed in the text and their inclusion in Figure 1 would make it large and confusing. Selected examples of risk factors and biomarkers are not shown in Figure 1 in order to avoid unjustified preference/suppression of certain risk factors and biomarkers at the expense of others. For this reason, Figure 2 with specific examples of risk factors and biomarkers has been deleted. Information about individual risk factors and biomarkers is now only in the text.
- Brain insulin signaling pathway and the association with type 2 diabetes mellitus is not shown in Figure 1, as it is a risk factor that falls under the group "Environmental stressors (internal and external)" together with many other factors, as mentioned in the text (e.g., in subheadings 2.1.1. Environmental, 3.1. Amyloid hypothesis., and 3.2. Amyloid beta pathology). To emphasize the connection between insulin resistance and the development of AD, a paragraph was added to the subheading "3.9. Metabolic dysregulation", which summarizes the link between insulin resistance and neuroinflammation, production of reactive oxygen species, and mitochondrial dysfunction in Alzheimer's disease.
Point 4: Page 3, Line 90-105 can be minimized as much as possible because it covers mostly general explanations.
Response 4: Lines 90-105 were minimized.
Point 5: Page 4, Page 137-142, the whole paragraph should have a citation of the reference. The author should explain between the less significant and little significant. In both cases, NSAIDs are mentioned as little and less significantly correlated with AD development. Some of the behavioural aspects were also included in the correlation, such as agreeableness, conscientiousness, and openness. Secondly, how this information is useful for your review.
Response 5: The reference was used: “Bellou V et al. Systematic evaluation of the associations between environmental risk factors and dementia: An umbrella review of systematic reviews and meta-analyses. Alzheimer’s Dement. 2017;13(4):406-418.” For Alzheimer's disease, this study (Bellou et al., 2017) found level of evidence (i) “convincing”(>1000 cases, p<10-6, I2<50%, 95% PI excluding the null value, no small-study effects and excess significance bias) for late-life depression and. type 2 diabetes mellitus; (ii) “highly suggestive” (>1000 cases, p<10-6, largest study with a statistically significant effect) for cancer, depression at any age, physical activity; (iii) “suggestive” (>1000 cases, p<10-3) for aluminum, education, herpesviridae infection, low-frequency electromagnetic fields, NSAIDs; and (iv) “weak” (the rest associations with p<0.05) for alcohol drinking, dietary intake of vitamin C, dietary intake of vitamin E, chlamydia pneumoniae infection, spirochetal infection, midlife BMI, mild traumatic brain injury, statins, agreeableness, conscientiousness, neuroticism, openness, aspirin, non-aspirin NSAIDs, fish intake, stroke.
- The reference (Bellou et al., 2017) was moved from the first sentence of the paragraph to the end of the paragraph.
- Information on the statistical significance of pro was added to the text: “highly significant” (p<10-6)), “less significant” (p<10-3), and “weakly significant” (p<0.05).
- Information was added that “Weakly significant were also agreeableness, conscientiousness, and openness”.
- The usefulness of this information for this review lies in the identification of risk factors that are not directly related to the amyloid, tau, and mitochondrial hypothesis, but whose association with these hypotheses cannot be ruled out given the complexity of brain function.
Point 6: Mitochondrial dysfunction is well correlated with the onset/progression of neurodegenerative diseases such as AD and Parkinson’s disease (PD). Figure 3 is a hypothetical diagram proposing mitochondrial dysfunction gradually develops throughout the ageing process. The x-axis is unclear as it has not mentioned “years of age" or “birth to death” or any other meaning. Further, can this hypothetical diagram apply to PD?
Response 6:
- The figure has been modified: "AD onset trigger (about -20 years)", "AD symptoms (about 65 years of age)", and "AD dementia" are added on the time axis x. Clarifying information and the meaning of the x-axis were added to the legend to the figure.
- I hypothesize that if longitudinal measurements of specific biomarkers of Parkinson's disease (PD) confirm their occurrence years before the manifestation of PD symptoms and disease progression leading to PD dementia, then a similar hypothetical diagram can be applied to PD.
Point 7: Page-10 to 19, the information on neurodegeneration in AD is well available in other published materials and the content can be minimized with a specific focus on major areas.
Response 7: Chapters 3 (Neurodegeneration in Alzheimer's disease) (pages 10-19) and 4 (Hypotheses) were merged into one chapter 3 (Hypotheses of Alzheimer's disease). Redundant text has been removed and the paragraphs have been ordered so that the hypothesis is presented first and the molecular mechanisms later.
Point 8: The review is very long and should focus on the linkage between A-beta-Tau and mitochondrial dysfunction. The author is advised to minimize the content making it more specific to the title. Otherwise, the reader is lost in the middle in understanding other contents.
Response 8:
- Former subheading 4.4. (Linking the amyloid, tau, and the mitochondrial hypotheses) has been transformed into a separate chapter 4. (Linking the amyloid, tau, and the mitochondrial hypotheses of Alzheimer's disease) to highlight the main goal of the manuscript.
- The entire chapter 5 (Alzheimer's drugs and adjuvants) has been reworked into chapter 5 (Alzheimer's drugs). Redundant information was deleted, the chapter was supplemented and transformed into subheadings: 5.1. (Cellular drug targets), 5.2. (Approved drugs), 5.3. (Drug candidates), and 5.4 (Supplements).
Round 2
Reviewer 2 Report
The author has responded to the comments and the manuscript is revised accordingly. Now the manuscript may be accepted for publication.